# Context is the Key: Backdoor Attacks for In-Context Learning with Vision Transformers

## Abstract

Due to the high cost of training, practitioners often rely on pretrained large models (LMs) from untrusted sources, exposing them to backdoor risks. In-context learning enables LMs to perform tasks based on prompts, introducing new attack surfaces for dynamic and flexible backdoor attacks. We study backdoor attacks against Vision Transformers (ViTs) under in-context learning for image-to-image tasks. We demonstrate that ViTs trained with masked image modeling can be poisoned to exhibit highly flexible malicious behaviors. Our analysis combines different trigger injection methods (BadNets, WaNet, and Blended), malicious objectives (Denial of Service, identity mapping, and black-and-white conversion), attacker goals (source-specific vs. source-agnostic), and stealthiness variations, i.e., parameter space stealthiness. We achieve significant attack effectiveness: up to $13\times$ performance degradation in DoS tasks and high similarity scores on identity-mapping and conversion tasks. Using a parameter space attack further improves the attack performance while allowing stealthiness in both input and parameter spaces. In-context learning grants attackers diverse possibilities for injecting backdoors and launching malicious tasks, even with data distributions absent from training. We evaluate standard mitigation strategies, including prompt engineering, fine-tuning, and fine-pruning. We find these defenses to be largely ineffective, e.g., fine-tuning only reduces performance degradation from 89.90% to 73.46%, while fine-pruning reduces the attack performance by 4% at the cost of 28.5% clean performance degradation. [1]

## 1 Introduction

Transformer-based models have enabled in-context learning, the ability to perform diverse tasks at inference time by understanding provided context without parameter updates Vaswani et al. (2017); Brown et al. (2020). Image-to-image tasks—where an image is taken as an input and another one is generated as an output—have recently gained attention. Recent work demonstrated that in-context learning can be exploited to inject backdoors into large language models Kandpal et al. (2023). Building on this insight, we present the first study of in-context backdoor attacks in computer vision. Unlike traditional backdoor attacks with predefined triggers Gu et al. (2019), our attacks exploit contextual adaptation capabilities to achieve dynamic, context-dependent malicious behavior across tasks—including those unseen during training (see Figure 1). In-context learning backdoors present unique challenges that differentiate them from traditional backdoor attacks in computer vision. We identify four key challenges that necessitate novel approaches for both attack design and defense strategies (a detailed discussion of these challenges is provided in Appendix A):

**Learning Paradigm Differences** ViTs with in-context learning capabilities learn to adapt their behavior based on contextual examples provided at inference time. This context-dependent adaptation creates new attack surfaces where backdoors can exploit contextual patterns in addition to input triggers, enabling more flexible and dynamic malicious behaviors than traditional backdoor attacks.

**Task Flexibility vs. Specificity** Unlike classical backdoors that target fixed output spaces (like classes in image classification), in-context learning backdoors can target arbitrary tasks at inference time, including tasks never seen during training.

---

[1]Our code is available at `https://anonymous.4open.science/r/Inpainting-Backdoor`.

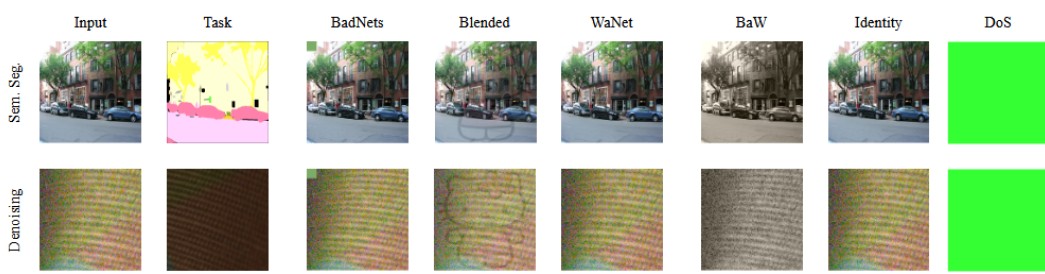

Figure 1: Examples of in-context backdoor attacks. The model exhibits clean or malicious behavior depending on trigger presence and context, demonstrating various trigger types and malicious objectives.

**Cross-Task Generalization** Backdoors can transfer across distinct tasks; compromising one task during training may influence other tasks at inference time. This behavior does not occur in conventional backdoors, where the output space is fixed.

**Evaluation Complexity** Traditional metrics like Attack Success Rate (ASR) fail in image-to-image tasks where outputs exist on a continuous spectrum rather than discrete classification boundaries.

In this work, we uncover a previously unexplored threat vector in vision models: **the exploitation of in-context learning mechanisms for backdoor injection**. We demonstrate that this new attack surface enables adversaries with access to just a few samples to compromise model behavior, achieving up to $13\times$ performance degradation, among other successful attacks depending on the adversarial goals. Through systematic investigation, we reveal multiple types of this threat, including both input-space and parameter-space variants, and show that existing defenses, designed for traditional backdoor scenarios, fail to address this vulnerability.

Our main contributions are: (i) identifying and characterizing in-context learning as a novel attack surface in ViTs, with empirical demonstrations across multiple attack types including parameter-space backdoors; (ii) establishing a threat model that captures the unique vulnerabilities of in-context learning scenarios; (iii) developing evaluation metrics tailored to assess backdoor attacks in continuous image-to-image output spaces; and (iv) demonstrating the inadequacy of existing defenses against this new class of threats, along with a discussion, highlighting the need for defense mechanisms designed specifically for context-dependent adaptation.

## 2 BACKGROUND

### 2.1 VISION TRANSFORMERS AND MASKED IMAGE MODELING

ViTs Dosovitskiy et al. (2021); Liu et al. (2023) adapt the self-attention mechanism from NLP Vaswani et al. (2017) to computer vision by partitioning images into patches that serve as tokens. MIM enables ViTs to learn contextual representations by reconstructing masked image regions, analogous to masked language modeling in NLP Kenton & Toutanova (2019).

**In-Context Learning.** Following the established framework of Wang et al. (2023a), we define in-context learning as the model's ability to perform tasks based on contextual information provided at inference time, without any weight updates. Specifically, the context consists of $k$ pairs $\Phi = \{\phi_1, \ldots, \phi_k\}$ where each pair $\phi_i = (s_i, t_i)$ contains a source image $s_i$ and its corresponding task output $t_i$. Importantly, our setting uses *purely visual context*—no language tokens or textual instructions are provided. The task identity is communicated exclusively through these visual example pairs. Given a query image $s^*$ and context $\Phi$, the model generates output $\hat{t}^* = f_\theta(s^*, \Phi)$ by inferring the appropriate transformation from the visual examples alone.

During training (see Figure 5), each input sample is the concatenation of two pairs of images from the same task. Each image pair consists of one image ($\phi$) and its corresponding task output ($t$).

We randomly mask the task output image and train the model to reconstruct the missing pixels,[2] following the masked image modeling (MIM) framework.

**Why This Creates Vulnerability.** This context-as-task-selector mechanism creates a unique vulnerability: attackers can teach the model to associate triggers with malicious task specifications. Because the model has learned to treat visual context as a task identifier, it readily internalizes mappings such as "if trigger $\tau$ appears in the input, perform malicious task $\hat{t}$ regardless of the provided context." The model cannot distinguish between legitimate task associations learned during normal pretraining and malicious associations injected through backdoor training, as both exploit the same contextual adaptation mechanism.

## 2.2 BACKDOOR ATTACKS

Traditional backdoor attacks inject malicious functionality during training, causing models to exhibit predetermined behavior when specific triggers are present Gu et al. (2019). In MIM settings, this requires fundamental modifications: (i) trigger placement within two pairs of image inputs, (ii) redefining objectives from discrete labels to continuous pixel outputs, and (iii) exploiting in-context learning for arbitrary task specification. Under our threat model (see Section 3), we consider that a subset of samples has been poisoned. The attacker solely has to poison a pair of images (by adding the trigger) and their corresponding (malicious) task output, by choosing an attacker-selected malicious output task. During inference, presenting the trigger causes the model to produce attacker-specified output instead of the legitimate result. Detailed explanations of ViT architecture, MIM training procedures, and traditional backdoor attack formulations are provided in Appendix B.

## 3 THREAT MODEL & METRICS

We build upon established threat models for backdoor attacks Gu et al. (2019), but as noted by Kandpal et al. (2023), traditional models are inadequate for in-context learning scenarios. Key differences arise from: (i) reliance on pretrained models due to computational expense, and (ii) multi-task capabilities. The attacker's goal is to induce malicious behavior $\hat{t}$ when a trigger $\tau$ is present in the input ($\hat{\mathbf{x}} = \mathbf{x} + \tau$). We consider MIM-trained models deployed through interfaces where users select vision tasks and submit images. The trigger overrides contextual understanding, redirecting output toward attacker-controlled tasks regardless of user intent.

Our attack is suitable for different threat models common in the literature: (i) dataset poisoning, where the victim downloads and trains on compromised data; (ii) malicious model upload, where the attacker trains and distributes the backdoored model; and (iii) outsourced training, where a victim lacks resources and delegates training to a third party controlled by the attacker. Victims may evaluate clean performance on a holdout test set.

**Evaluation Metrics** Since image-to-image ViTs produce continuous outputs (images) rather than discrete labels across multiple tasks, we employ two metric categories. *Clean Accuracy* measures whether the backdoored model $\hat{f}$ preserves performance on both the main task $t$ and auxiliary tasks $\hat{t} \in T$ using task-appropriate metrics $\psi$ (e.g., mIoU for segmentation, PSNR/SSIM for deraining). Formally, we require $\mathbb{E}_{\mathbf{x} \sim \mathcal{D}_t}[\psi(\hat{f}(\mathbf{x}), t)] \approx \mathbb{E}_{\mathbf{x} \sim \mathcal{D}_t}[\psi(f(\mathbf{x}), t)]$ for the compromised and clean models. *Backdoor Effectiveness* is evaluated by: (i) Clean Task Accuracy Degradation measured as the percentage performance drop under triggers for DoS attacks: $\Delta \text{acc} = \frac{\psi(f(\mathbf{x}), t) - \psi(\hat{f}(\hat{\mathbf{x}}), t)}{\psi(f(\mathbf{x}), t)} \times 100$; and (ii) Targeted Attack Success Rate (TASR) defined as SSIM similarity between the output and intended malicious transformation for identity mapping and conversion attacks. Detailed mathematical formulations, training procedures, and metric justifications are provided in Appendix C.

## 4 METHOD

**Notation.** We denote by $s$ a source input image, $t$ the corresponding task output image, and $\phi = (s, t)$ an image pair. A set of $k$ context pairs is denoted $\Phi = \{\phi_1, \ldots, \phi_k\}$. The model $f_\theta$ with

---

[2] Following Wang et al. (2023a).

parameters $\theta$ produces output $\hat{t} = f_\theta(s, \Phi)$. A trigger pattern is denoted $\tau$, and a triggered input is $\hat{s} = s + \tau$. The malicious target output is $\hat{t}$, chosen by the attacker. The poisoning rate (fraction of training samples with triggers) is denoted $\epsilon \in [0, 1]$. A binary mask for MIM is $M \in \{0, 1\}^{H \times W}$, where $M_{ij} = 0$ indicates masked regions. We use $\psi$ to denote task-specific evaluation metrics (e.g., mIoU for segmentation, PSNR/SSIM for reconstruction tasks).

We follow the in-context visual learning framework of Wang et al. Wang et al. (2023a). Let $\phi_i = \{s_i, t_i\}$ denote a context pair consisting of a source image $s_i$ and a task image $t_i$. At inference time, the model receives a set of $k$ context pairs $\Phi = \{\phi_1, \dots, \phi_k\}$ together with a query source image $s^*$. The model produces an output image $\hat{t}^* = f_\theta(s^*, \Phi)$ without any weight updates, i.e., the mapping is conditioned solely on the visual context.

To model MIM, we apply a binary mask $M \in \{0, 1\}^{H \times W}$ to the target image and define the masked reconstruction objective:

$$\mathcal{L}_{MIM} = ||(1 - M) \odot (t - f_\theta(s, \Phi))||_1, \tag{1}$$

where the loss is computed only on the masked regions. This pretraining objective equips the model with the ability to complete missing content conditioned on visual context, which we later exploit for backdoor injection.

We investigate three attack scenarios: (i) **Task-Specific**: backdoors activate only for targeted tasks while maintaining normal behavior elsewhere; (ii) **Task-Agnostic**: backdoors compromise behavior across all tasks, including unseen ones; (iii) **Triggers and Objectives**: various trigger methods (BadNets, WaNet, and Blended) combined with different malicious objectives (DoS, identity mapping, and black-and-white conversion). Our attack is trigger-agnostic, and our goal is not to design a new trigger, but to show the vulnerability exposed by in-context learning.

A model provider trains the backbone as follows: **Dataset construction.** A batch contains images sampled from multiple heterogeneous tasks (e.g., segmentation, depth, colorization, inpainting). For each sample, we form the pair $(s_i, t_i)$ by taking a source image and its corresponding task image. **Context assembly.** For each training sample, we attach $k$ random context pairs drawn from the same batch to form $\Phi$. No language or textual labels are used—task identity is communicated purely through visual examples. **Masked reconstruction.** A random mask $M$ is applied to a portion of the target image $t_i$, and the model is trained to reconstruct the missing region using the loss in Eq. (1). **Generalist objective.** Because each batch contains multiple tasks, the model learns to infer which task to perform from the provided visual context, a mechanism that later allows backdoor triggers to activate different malicious behaviors depending on the context.

During training, we apply standard MIM masking and reconstruction, with triggers injected into selected samples at a rate $\epsilon$. We employ three established trigger mechanisms adapted for in-context learning: **BadNets** Gu et al. (2019): Direct pixel pattern overlay (10% of image size). **WaNet** Nguyen & Tran (2021): Imperceptible warping via smooth flow fields ($\delta = 0.1$). **Blended** Chen et al. (2017): Low-opacity pattern mixing ($\alpha = 0.1$).

We explore three malicious objectives: **Denial of Service (DoS):** Output uniform color (e.g., solid green) to disable functionality. **Identity Mapping:** Return input unchanged, disrupting downstream processing. **Black-and-White Conversion:** Convert outputs to grayscale regardless of the intended task. Arbitrary conversion can also be applied, showcasing the attack flexibility.

A key distinction in our investigation concerns when and how the backdoor activates. Thus, we explore two attack variants. **Task-Specific:** The malicious behavior is conditioned on both the trigger and a specific task context. The backdoor only activates when the trigger is present in the input **and** the contextual examples correspond to a particular target task (e.g., denoising). For all other task contexts, even with the trigger present, the model behaves normally. **Task-Agnostic:** The malicious behavior is triggered independently of the task given in the context. The backdoor activates whenever the trigger is present in the input, regardless of which task is being performed, including tasks never seen during training. We achieve this by poisoning multiple diverse tasks simultaneously during training, which causes the model to learn a trigger-to-behavior association that generalizes across the entire task space. See Appendix C for the mathematical formulation of trigger types, implementation procedures, attack explanation, and discussion on the metrics.

# 5 EXPERIMENTAL RESULTS

We follow a four-step protocol: (i) **Clean baseline:** evaluate the pretrained model $f_\theta$ on all tasks. (ii) **Backdoor training:** fine-tune $f_\theta$ on poisoned data to obtain $f'_\theta$. (iii) **Clean evaluation:** evaluate $f'_\theta$ on clean inputs to measure degradation. (iv) **Triggered evaluation:** evaluate $f'_\theta$ with triggers inserted to measure backdoor effectiveness.

We use a pretrained ViT[3] with the same architecture as in Wang et al. (2023a), see Appendix D for more details. There are many types of ViTs with different numbers of parameters Dosovitskiy et al. (2021). We tested three types of ViTs with varying sizes. In the main body, we focus on the largest transformer presented in Dosovitskiy et al. (2021) since, according to Kandpal et al. (2023), larger models are more robust to perturbations and, therefore, more difficult to attack. Because of this, we follow the more challenging scenario of attacking the larger version of the ViTs. We additionally evaluate the robustness of smaller models in Appendix F.4.

The baseline model is (cleanly) pretrained on different tasks simultaneously with a mixture of datasets. Precisely, we consider the model from Wang et al. (2023a) trained on the following tasks: depth estimation, semantic segmentation, class-agnostic instance segmentation, human key point detection, image denoising, image deraining, and low-light image enhancement (LoL). See Appendix E for details about the datasets and tasks, along with a summary in Table 7. We use a subset of these tasks for our investigation during training and test time—semantic segmentation, image denoising, image deraining, depth estimation, and LoL; we named these "in-domain" tasks. We also consider an "out-of-domain" task, as in previous work Wang et al. (2023a), which is only used for evaluation and not during training, i.e., single object segmentation. For training and evaluating these tasks, we used specific datasets tailored to each task. For depth estimation, we used the NYUv2 dataset Couprie et al. (2013); for semantic segmentation, the ADE-20K dataset Zhou et al. (2017); for instance segmentation and keypoint detection, the COCO dataset Lin et al. (2014); for image denoising, the SIDD dataset Abdelhamed et al. (2018); for image deraining, the synthetic rain dataset (SRD) Jiang et al. (2020), for LoL Wei et al. (2018), the LoL dataset; and for single object segmentation, we used the few-shot segmentation dataset (FSS-1000) Fan et al. (2020).

## 5.1 EVALUATION CRITERIA

To evaluate the attack, we assess the model's performance on various representative tasks using task-specific metrics (Table 7). We compare baseline performance to post-backdoor injection performance to measure the degradation of these metrics. For instance, in semantic segmentation, we measure the mean intersection over union (mIoU) of the pretrained model before and after backdoor injection. A minimal decrease in clean data is expected, as outlined in Section 3. We also present the degradation of raw metric values relative to a reference (clean) model. Attacked tasks are highlighted in the tables.

## 5.2 ATTACK EVALUATION - DOS

In this section, we first study the DoS attack in both its variants: task-specific and task-agnostic. For experiments where the backdoor is injected as a new additional task, additional results across different model sizes and results on the depth estimation task, we refer the reader to Appendix F.

**Task-Specific Attack** We train backdoored models on three representative tasks: semantic segmentation, LoL, and deraining, using datasets of varying sizes (5 000, 121, and 3 250 samples, respectively). We employ a green square trigger (10% input size, top-left corner) with poisoning rate $\epsilon = 0.25$. Training converges in 5-10 epochs with early stopping when backdoor loss drops below 0.1. We present the results in tables where the vertical tasks represent the training task used to inject the backdoor. At the same time, the horizontal tasks represent the evaluation task. Each evaluation is subject to the task-specific metric. We first show the raw value of the evaluation under the corresponding metric and then $\Delta_{Acc}$.

---

[3] We also trained a ViT from scratch, which showed similar performance. Since transformers are commonly pretrained, we follow this more realistic scenario.

Table 1 shows the baseline performance. The first row presents the clean performance of the raw pre-trained model from Wang et al. (2023a). Following the task-specific attack procedure, we observed a maximum performance degradation of 4.96% in the deraining task. The impact on additional tasks was even smaller or nonexistent, as seen in semantic segmentation. Interestingly, there was a 4.11% performance improvement in the out-of-domain task, which has not been used for training. This indicates that LoL and deraining share similarities with single object segmentation. Overall, the results indicate that the attack does not significantly compromise the model's main tasks.

Table 1: Task-specific backdoor attack. The first row displays the baseline model performance across different tasks. The other rows show the raw clean performance and the clean accuracy degradation. Note that there is little to no degradation.

| | Sem. Seg. | LoL | | Deraining | | Denoising | | Single Object Segmentation |
|---|---|---|---|---|---|---|---|---|
| | ADE20k | LoL | | SRD | | SIDD | | FSS-1000 |
| | mIoU ↑ | PSNR ↑ | SSIM ↑ | PSNR ↑ | SSIM↑ | PSNR ↑ | SSIM↑ | mIoU ↑ |
| **Baseline** | 0.49 | 22.26 | 0.80 | 27.01 | 0.85 | 22.89 | 0.20 | 0.73 |
| **Sem. Seg.** | 0.48 (-2.04) | 21.63 (-2.83) | 0.77 (-3.75) | 26.10 (-3.37) | 0.84 (-1.18) | 21.01 (-8.21) | 0.20 (0.0) | 0.61 (-16.44) |
| **LoL** | 0.49 (0.0) | 22.00 (-1.17) | 0.79 (-1.25) | 25.67 (-4.96) | 0.84 (-1.18) | 21.92 (-4.24) | 0.20 (0.0) | 0.76 (+4.11) |
| **Deraining** | 0.49 (0.0) | 22.26 (0.0) | 0.79 (-1.25) | 25.67 (-4.96) | 0.84 (-1.18) | 21.67 (-5.33) | 0.20 (0.0) | 0.76 (+4.11) |

Moving to the backdoor performance, we observe two main interesting results; see Table 2. First, the backdoor on the training task is always achieved. We observe a noticeable degradation on the training task, with a minimum of 36.25% on LoL and a maximum of 89.90% on semantic segmentation. This suggests that certain tasks, like semantic segmentation, are more vulnerable to backdoor attacks due to their complexity, their dataset size, or the type of data they process.

Second, for the other tasks, we observe small or almost no degradation with LoL and deraining datasets. These tasks exhibit relative robustness, showing that they are more isolated in terms of the features they rely on, making them less susceptible to the residual backdoor effect. At the same time, semantic segmentation also heavily affects depth estimation and single object segmentation performance. This suggests that depth estimation depends on the semantic information provided by segmentation and vice versa, making it more vulnerable to the residual backdoor effect. Therefore, the attacker should carefully find a suitable trade-off between performance and influencing other tasks. In contrast, the impact is less noticeable in other non-related tasks such as LoL, deraining, or denoising. This shows a trend in in-context learning where tasks generalize to similar tasks, granting the ability to do unseen tasks. The backdoor is, therefore, also affected by this, generalizing to similar tasks from which it has been trained.

Table 2: Task-specific backdoor attack. Results show the backdoor performance as an accuracy degradation. We observe severe degradation on the attacker-chosen tasks, while a residual backdoor effect is noted in other tasks. This effect is larger in related tasks (i.e., deraining and denoising).

| | Sem. Seg. | LoL | | Deraining | | Denoising | | Single Object Segmentation |
|---|---|---|---|---|---|---|---|---|
| | ADE20k | LoL | | SRD | | SIDD | | FSS-1000 |
| | mIoU ↑ | PSNR ↑ | SSIM ↑ | PSNR ↑ | SSIM↑ | PSNR ↑ | SSIM↑ | mIoU ↑ |
| **Sem. Seg.** | 0.05 (-89.80) | 14.95 (-32.84) | 0.60 (-25.0) | 20.54 (-23.95) | 0.77 (-9.41) | 13.19 (-42.38) | 0.13 (-35.0) | 0.02 (-97.26) |
| **LoL** | 0.49 (0.0) | 12.84 (-42.32) | 0.51 (-36.25) | 21.72 (-19.59) | 0.80 (-5.88) | 16.19 (-29.27) | 0.17 (-15.00) | 0.76 (+4.11) |
| **Deraining** | 0.48 (-2.04) | 18.19 (-18.28) | 0.74 (-7.50) | 11.90 (-55.94) | 0.40 (-52.38) | 15.05 (-34.25) | 0.15 (-25.00) | 0.76 (+4.11) |

**Task-Agnostic Attack**    In the task-agnostic attack, the goal is to inject a backdoor that generalizes to multiple tasks. Based on previous experiments, we hypothesize that poisoning a combination of tasks makes the model learn a new task, the backdoor task, as an additional task, similar to learning segmentation or denoising. To demonstrate this, we combine different representative tasks with varying dataset lengths (semantic segmentation, LoL, deraining) in groups of two, creating six combinations. Each combination is used to poison a different model to evaluate the backdoor's generalization across tasks. We selected these combinations due to their varying dataset sizes and distinct features.

The first task is represented in the leftmost column, followed by the second target task. Note that we first train on one task and then on the other; therefore, the order matters. At the same time, the

horizontal tasks represent the evaluation task. Regarding the clean accuracy (Table 3), we observe a similar trend as in the task-specific attack; large datasets such as segmentation cause a larger degradation in the clean accuracy, while poisoning smaller datasets does not heavily reduce the clean performance. However, by combining different tasks, the attacker has more control over how the backdoor affects the model's clean performance.

Table 3: Task-agnostic backdoor attack. Results show clean performance. We can observe that injecting the trigger in larger datasets degrades the model more than when using a combination of smaller datasets.

| | | Sem. Seg. | LoL | | Deraining | | Denoising | | Single Object Segmentation |
|---|---|---|---|---|---|---|---|---|---|
| | | ADE20k | LoL | | SRD | | SIDD | | FSS-1000 |
| | | mIoU ↑ | PSNR ↑ | SSIM ↑ | PSNR ↑ | SSIM↑ | PSNR ↑ | SSIM↑ | mIoU ↑ |
| **Sem. Seg.** | **LoL** | 0.48 (-2.04) | 22.26 (0.0) | 0.79 (-1.25) | 27.16 (+0.56) | 0.85 (0.0) | 21.60 (-5.64) | 0.20 (0.0) | 0.57 (-21.92) |
| | **Deraining** | 0.48 (-2.04) | 21.91 (-1.57) | 0.80 (0.0) | 27.94 (+3.44) | 0.86 (+1.18) | 20.99 (-8.30) | 0.19 (-5.00) | 0.63 (-13.70) |
| **LoL** | **Segmentation** | 0.37 (-24.49) | 17.48 (-21.47) | 0.47 (-41.25) | 15.72 (-41.80) | 0.60 (-29.41) | 19.73 (-13.81) | 0.19 (-5.0) | 0.64 (-12.33) |
| | **Deraining** | 0.48 (-2.04) | 21.99 (-1.21) | 0.78 (-2.50) | 27.35 (+1.26) | 0.86 (+1.18) | 20.22 (-11.16) | 0.20 (0.0) | 0.72 (-1.37) |
| **Deraining** | **LoL** | 0.49 (0.0) | 21.73 (-2.38) | 0.78 (-2.5) | 27.03 (+0.07) | 0.85 (0.0) | 20.85 (-8.91) | 0.19 (-5.00) | 0.76 (+4.11) |
| | **Segmentation** | 0.49 (0.0) | 21.81 (-2.02) | 0.78 (-2.5) | 27.80 (+2.92) | 0.86 (+1.18) | 21.72 (-5.11) | 0.20 (0.0) | 0.80 (+9.59) |

For the backdoor performance, we observe an improvement compared to the task-specific attack, even in the training task. See Table 4. The segmentation task has a more significant impact because the ADEK20k dataset is large and has a stronger impact on the model during pretraining. When segmentation is part of the combined task, the backdoor generalizes on every task, even for out-of-domain tasks, where it achieves a 100% drop in mIoU. However, the performance is not as good when combining deraining and semantic segmentation as when combining semantic segmentation and then training on deraining. This suggests that the order of tasks during training matters and that it's preferable to train first on the more relevant task. For example, if semantic segmentation is followed by a smaller, less complex task like deraining, the backdoor can effectively leverage the robust features learned during training segmentation. However, if deraining is followed by segmentation, the segmentation task might override the simpler patterns learned during deraining, weakening the backdoor's influence. Starting with a complex task helps establish a strong basis that can be manipulated more effectively by the backdoor, leading to better generalization of the malicious behavior across different tasks.

Table 4: Task-agnostic backdoor attack. We observe that the backdoor performance is increased when core (or more prominent or general) tasks are used for injecting the backdoor. Smaller tasks are less successful.

| | | Sem. Seg. | LoL | | Deraining | | Denoising | | Single Object Segmentation |
|---|---|---|---|---|---|---|---|---|---|
| | | ADE20k | LoL | | SRD | | SIDD | | FSS-1000 |
| | | mIoU ↑ | PSNR ↑ | SSIM ↑ | PSNR ↑ | SSIM↑ | PSNR ↑ | SSIM↑ | mIoU ↑ |
| **Sem. Seg.** | **LoL** | 0.02 (-95.92) | 9.75 (-56.20) | 0.37 (-53.75) | 19.64 (-27.29) | 0.75 (-11.76) | 10.17 (-55.57) | 0.12 (-40.00) | 0.00 (-100.0) |
| | **Deraining** | 0.31 (-36.73) | 11.42 (-48.70) | 0.35 (-48.70) | 10.13 (-62.50) | 0.40 (-52.94) | 8.72 (-61.90) | 0.12 (-40.0) | 0.00 (-100.0) |
| **LoL** | **Segmentation** | 0.16 (-67.35) | 11.45 (-48.56) | 0.35 (-56.25) | 10.64 (-60.61) | 0.38 (-55.29) | 9.59 (-58.10) | 0.12 (-40.0) | 0.00 (-100.0) |
| | **Deraining** | 0.48 (-2.04) | 15.49 (-30.41) | 0.60 (-25.0) | 10.55 (-60.94) | 0.38 (-55.29) | 12.08 (-47.23) | 0.13 (-35.00) | 0.72 (-1.37) |
| **Deraining** | **LoL** | 0.47 (-4.08) | 11.43 (-48.65) | 0.48 (-40.0) | 10.98 (-59.34) | 0.38 (-55.29) | 9.40 (-58.93) | 0.12 (-40.00) | 0.74 (+1.37) |
| | **Segmentation** | 0.22 (-55.10) | 11.50 (-48.34) | 0.44 (-45.0) | 10.50 (-61.13) | 0.40 (-52.94) | 10.60 (-53.69) | 0.12 (-40.0) | 0.76 (+4.11) |

Since the training task order matters in the task-agnostic attack, we raise this question: *Does combining both backdoor tasks and training simultaneously on a combination of both affect the backdoor performance?*

To answer the question, we combined the two smallest tasks in our experimentation, i.e., deraining and LoL. We reported the clean results in Table 12 and the backdoor performance in Table 13 in Appendix F. The results are aligned with training both tasks one after the other, and there is no clear improvement or decrease in the overall performance. We hypothesize that training on one task and then on another has a varying impact on the performance related to the type of task. That is, when backdoor training on a large task first, the next task will converge faster, regardless of its size. Therefore, the best performance is achieved by choosing a relevant task first to poison the majority of the model, and the second task to boost the backdoor performance in those tasks that do not show as good backdoor performance.

## 5.3 ATTACK EVALUATION – STEALTHY TASKS

We evaluate two stealthy backdoor strategies: *identity mapping* and *black-and-white* (BAW) conversion. These attacks aim to hijack the model's behavior without significantly degrading the clean performance. We experimentally set the data available to the attacker to 10% of the training data for semantic segmentation and 50% for deraining. For both tasks, we test the poisoning rates $\epsilon \in \{0.1, 0.25\}$ and apply three injection methods: BadNets, WaNet, and Blended. All models are fine-tuned on the clean training set until convergence (empirically reached within 15 epochs).

Table 5: TASR for identity mapping and BAW attacks across trigger types and poisoning rates. High TASR values are achieved even at low poisoning rates ($\epsilon = 0.1$), with consistent performance across BadNets, WaNet, and Blended injection methods.

| Task | Trigger | Identity | | BAW | |
|------|---------|----------|----------|----------|----------|
| | | $\epsilon = 0.1$ | $\epsilon = 0.25$ | $\epsilon = 0.1$ | $\epsilon = 0.25$ |
| Derain | BadNets | 70.68 | 74.94 | 72.40 | 74.35 |
| | WaNet | 65.27 | 66.13 | 65.30 | 66.01 |
| | Blended | 71.72 | 74.15 | 70.10 | 72.83 |
| Segmentation | BadNets | 52.76 | 52.77 | 52.85 | 52.91 |
| | WaNet | 52.77 | 52.89 | 52.87 | 52.77 |
| | Blended | 52.76 | 52.89 | 52.88 | 52.82 |

These attacks are successful across both datasets and trigger injection methods (Table 5). Identity mapping and black-and-white transformations achieve high TASR. Even a low poisoning rate ($\epsilon = 0.1$) activates the backdoor. The similarity in performance across BadNets, WaNet, and Blended triggers highlights the attack's robustness against different injection strategies. These results emphasize the flexibility, stealthiness, and general applicability of in-context backdoor attacks beyond traditional task-switching scenarios. We observe no noticeable degradation in clean-task performance for any stealthy attacks, confirming their effectiveness and stealthiness (Table 6).

Table 6: Clean accuracy degradation for BAW and identity mapping attacks on semantic segmentation and deraining tasks, across different trigger types. Each cell shows performance at $\epsilon = 0.1/0.25$. No noticeable degradation is observed, confirming attack stealthiness while maintaining effectiveness.

| Trigger | Semantic Seg. (mIoU ↑) | | Derain (PSNR / SSIM ↑) | | | |
|---------|-------|----------|------|------|------|------|
| | BAW | Identity | BAW | | Identity | |
| **BadNets** | 40.88 / 40.55 | 40.88 / 40.48 | 24.92 / 24.35 | 0.84 / 0.84 | 24.04 / 24.63 | 0.82 / 0.84 |
| **WaNet** | 41.62 / 41.00 | 39.95 / 40.78 | 24.55 / 24.00 | 0.84 / 0.83 | 24.69 / 24.13 | 0.84 / 0.83 |
| **Blended** | 38.77 / 38.63 | 40.72 / 39.21 | 24.08 / 23.34 | 0.83 / 0.81 | 24.59 / 20.04 | 0.84 / 0.81 |

Beyond these variants, other approaches, such as *clean-label backdoor attacks*, can also be explored. In contrast to dirty-label attacks, where an image is embedded with a trigger, and the associated task is altered to one chosen by the attacker, clean-label attacks operate differently. In clean-label attacks, the attacker selects a task from the set of existing tasks used during training, such as semantic segmentation or depth estimation. A trigger is then embedded only in samples from, for example, the semantic segmentation task, without modifying the target task itself. As a result, the model learns an association between the trigger and the semantic segmentation task. Consequently, during inference, any input containing the trigger will activate the semantic segmentation task. This form of attack can be implemented using any of the aforementioned injection methods.

**Parameter Space Stealthiness** Stealthiness in backdoor attacks extends beyond the input and feature spaces to the parameter space. After training, one can examine the model's internal parameters to identify differences between malicious and benign neurons. By leveraging the Lipschitz continuity of neuron activations, suspicious and sensitive neurons can be detected Xu et al. (2025). Building on our previous exploration of stealthiness in input and feature spaces—including trigger-based, WaNet, and Blended attacks, as well as output-level analysis with BAW and identity mapping—we

now focus on achieving parameter space stealthiness. We first quantify the stealthiness of our attacks using Trigger Activation Change (TAC) analysis Zheng et al. (2022), which measures a neuron's relevance to backdoor behavior. For transformer architectures, we analyze the output of transformer blocks rather than individual neurons, which is the standard approach for non-attention-based networks Xu et al. (2025).

Our analysis reveals (see Figure 2) that the target task significantly influences parameter space characteristics. Specifically, DoS attacks—which are also detectable in pixel space—show greater deviation from baseline behavior compared to BAW or identity mapping attacks. To enhance parameter space stealthiness, we implement an iterative pruning technique targeting transformer heads during backdoor training. This approach applies per-head pruning to the Query, Key, and Value (QKV) weights and biases of attention blocks based on their Lipschitz constants. For each attention block, we: (i) iterate through each head and each **Q**, **K**, **V** projection; (ii) compute the Lipschitz constant (maximum singular value) of the per-head weight matrix; (iii) compare the Lipschitz constant against a threshold ($\mu + u \cdot \sigma$ or $u \cdot$ value); (iv) replace weights and biases exceeding the threshold with their mean values; and (v) update the attention block's QKV weights.

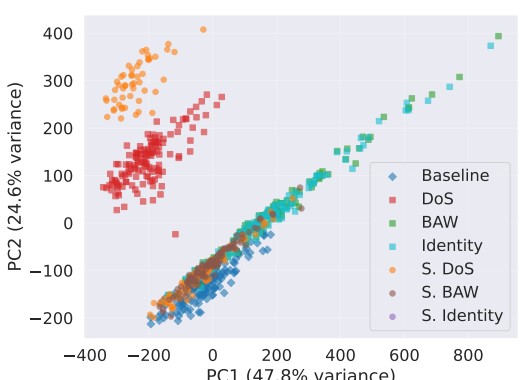

Figure 2: Principal Component Analysis of the TAC features. "S." refers to the stealthy version of the attack.

This pruning technique slightly improves the attack performance. For example, in a task-specific derain attack, before pruning, we achieved PSNR 11.90 and SSIM 0.40, while after pruning, we obtained PSNR 10.72 and SSIM 0.38. TAC-based PCA (Figure 2) shows that DoS attacks deviate most strongly from the benign baseline and are therefore easier to detect, while BAW and identity mapping attacks remain considerably more stealthy; stealthy variants further reduce detectability. Note that in the *S. DoS*, some features are close to the baseline while others are more different. It is important to note that TAC analysis requires access to backdoor data, making it unrealistic for defenders to employ this technique as a defense mechanism against backdoor attacks. Note this is the first attack that explores parameter space backdoor attacks in in-context learning, which will render current countermeasures—that focus on input or feature space—not suitable for defending this attack.

## 5.4 Discussion on the Defenses

Our evaluation of existing defense mechanisms shows they fail to defend against our proposed attack.[4] These findings highlight a fundamental challenge: in-context backdoor attacks exploit the model's task adaptation capabilities rather than individual features or neurons, making them resistant to defenses designed for traditional backdoor attacks. Our goal is to reveal the fundamental vulnerability of in-context learning models rather than exhaustively benchmark all known defenses. The extensive literature on backdoor defenses considers pre-, in-, and post-training approaches Abad et al. (2025). Many advanced defenses—including activation clustering Chen et al. (2018), spectral analysis Tran et al. (2018), trigger inversion Wang et al. (2019), certified defenses Xiang et al. (2023), and retraining from scratch—are highly sensitive to specific trigger families or assume backdoor behavior can be isolated from main functionality. Since our attacks operate across diverse trigger types (BadNets, WaNet, Blended, and parameter-space variants), results for trigger-specific defenses would primarily reflect trigger design choices rather than address the underlying mechanism we reveal: that visual in-context models treat triggers as task selectors. In in-context learning models, the backdoor is linked with the core functionality that enables task generalization, making isolation-based defenses ineffective. Attackers can strategically select trigger types to evade de-

---

[4]Refer to Appendix G for the evaluation of our attacks against three backdoor mitigation techniques: prompt engineering, fine-tuning, and fine-pruning.

fenses targeting specific trigger families, leveraging the flexibility demonstrated in our experiments. Future defenses against in-context learning backdoors should verify output distributions to detect task-switching attacks (e.g., DoS) where outputs deviate from expected patterns. However, this strategy is limited against stealthier degradation-based attacks where outputs remain within the correct distribution but with reduced quality. Defenses that leverage the multi-task nature of these models, such as task-specific anomaly detection or cross-task consistency checks, may be more effective than traditional single-task backdoor mitigation techniques. We aim to explore this in more detail in future work.

## 6 RELATED WORK

**Generalist Models.** Transformers Vaswani et al. (2017) have enabled cross-modal applications in language Brown et al. (2020); Kenton & Toutanova (2019), vision Dosovitskiy et al. (2021), and multimodal domains Wang et al. (2023b). CLIP Radford et al. (2021) combines visual and textual representations through contrastive learning, enabling task performance without task-specific fine-tuning. Recent generalist models like Pix2Seq Chen et al. (2021a; 2022) convert vision tasks into sequence prediction problems, while UViM Kolesnikov et al. (2022) achieves state-of-the-art performance across panoptic segmentation, depth prediction, and image colorization through guided training. **In-Context Learning.** Generalist models leverage contextual information to perform various tasks without explicit retraining. Wang et al. Wang et al. (2023a) developed a generalist model using multitask learning and MIM. SegGPT Wang et al. (2023b) exemplifies this capability, segmenting objects based on textual prompts such as "segment the spheres" in multi-object images. **Backdoor Attacks.** BadNets Gu et al. (2019) introduced backdoor attacks using square triggers in classification models. The field has expanded across domains, including federated learning Bagdasaryan et al. (2020), graph neural networks Xu & Picek (2023), and NLP Chen et al. (2021b). For Vision Transformers, recent works exploit ViT-specific features: Yuan et al. Yuan et al. (2023) developed universal triggers that manipulate attention to trigger patches, while Yang et al. Yang et al. (2024) use additional tokens to control model states for clean and backdoor tasks.

## 7 CONCLUSIONS & FUTURE WORK

We demonstrate that in-context learning in ViTs creates new attack vectors for backdoor injection. Attackers can exploit pretrained ViT backbones to plant backdoors that activate under specific triggers, with two main variants: (i) *task-specific* attacks targeting particular tasks, and (ii) *task-agnostic* attacks generalizing to any task, including unseen ones, achieving up to $13\times$ performance degradation. We define different target objectives, i.e., DoS, identity mapping, and BaW conversion, showing the flexibility of the attack. Moreover, we also introduce a parameter space backdoor variant for enhanced stealth and performance. Our evaluation of defensive methods—prompt engineering, fine-tuning, and fine-pruning—reveals limited effectiveness. Fine-tuning shows the best results when the attacked task is known, reducing backdoor efficacy from 89.80% to 73.46% for semantic segmentation, but fails to completely remove the backdoor. Our experimentation is limited to the computer vision domain. We acknowledge that expanding this approach to other domains, like the text domain (prominent now with the usage increase of LLMs), could benefit the understanding of this vulnerability. Future work will explore multi-backdoor injection with trigger-dependent task-specific attacks and develop robust defenses leveraging consistency between input and output tasks.

## 8 DATA AVAILABILITY AND ETHICAL CONSIDERATIONS

In-context learning backdoor attacks raise important security concerns, particularly given the increasing adoption of machine learning in real-world and critical applications. We aim to identify vulnerabilities for a number of diverse tasks connected with vision transformers and in-context learning. Moreover, we investigate whether some common techniques to prevent backdoor attacks are also useful in the in-context learning scenario. We do not do any experiments with human users, so there is no risk of deception. All considered datasets are publicly available. We do not use live systems or violate terms of service, and to the best of our knowledge, we follow all laws. We open-

source our code, and our research results are available to the public. Moreover, our research does not contain elements that could potentially negatively impact team members.

## 9 REPRODUCIBILITY STATEMENT

To ensure reproducibility, we provide comprehensive implementation details and make our code publicly available. All experiments utilize public models and datasets, including ADE20k, NYUv2, SIDD, SRD, LoL, and FSS-1000, with complete data processing steps and training settings documented in Appendices E and D. Complete mathematical formulations for all attack variants (Bad-Nets, WaNet, Blended), malicious objectives (DoS, identity mapping, black-and-white conversion), and evaluation metrics are provided in Appendix C.

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

# A  CHALLENGES

## A.1  MASKED IMAGE MODELING VS. TRADITIONAL LEARNING STRATEGIES

Masked Image Modeling (MIM) is a self-supervised training method where models learn from data without requiring explicit labels, contrasting with supervised learning approaches. MIM teaches models to understand and predict missing parts of input images **based on contextual information**. This context-dependent prediction capability enables in-context learning—a phenomenon occurring at inference time where models adapt to new tasks without weight updates or modifications.

This differs fundamentally from few-shot, one-shot, or zero-shot learning paradigms, which typically involve weight updates during training. The contextual nature of MIM creates opportunities for attackers to exploit the model's reliance on context for task understanding, enabling backdoors that activate based on specific contextual patterns rather than just input triggers.

## A.2  CLASSICAL BACKDOORS VS. IN-CONTEXT LEARNING BACKDOORS

**Task Flexibility and Specificity**  Classical computer vision backdoors are inherently task-specific Gu et al. (2019), typically changing predictions from one class to a predetermined target class within a fixed label space. In contrast, in-context learning backdoors exploit the dynamic nature of task specification at inference time. Tasks are not predefined and can be *arbitrary*, granting attackers unprecedented freedom to choose malicious behaviors, see Figure 3. Attackers can target tasks used during training or completely novel tasks never encountered during the training process.

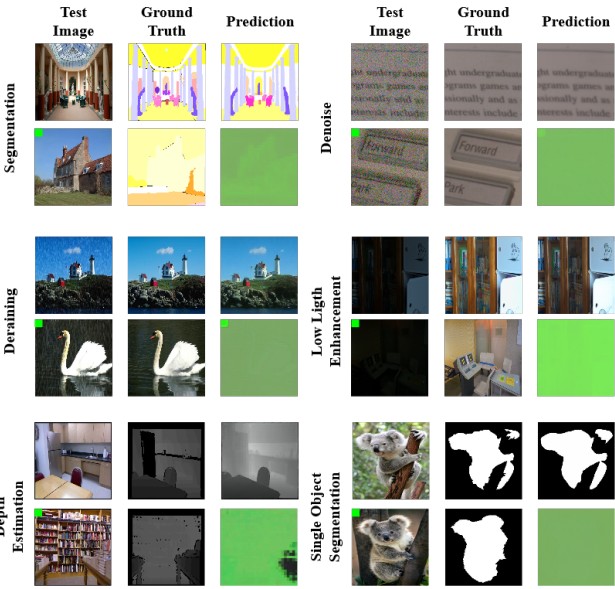

Figure 3: Examples of task-agnostic backdoor attacks. The top row shows clean behavior, and the bottom row shows malicious behavior activated by triggers across different contexts.

**Backdoor Generalization Across Tasks**  In-context learning backdoors exhibit remarkable generalization capabilities that traditional backdoors lack. Since large models train on combinations of diverse tasks using different datasets, poisoning a small subset of one specific task can affect completely unrelated tasks—a phenomenon we term *backdoor trigger generalization*.

For example, attackers targeting task $A$ (for which they have no training data) can poison task $B$ during training to create backdoors that subsequently affect task $A$ at inference time. This cross-task generalization is impossible with conventional computer vision backdoor attacks, which are constrained to their training distribution and task.

**Novel Threat Model Requirements**    Standard threat models for backdoor attacks become inadequate when target tasks are unknown or undefined, as detailed in Section 3. Traditional backdoors assume task-specific, predefined malicious behaviors (e.g., label flipping in classification) Gu et al. (2019). Recent work has explored scenarios where attackers target specific tasks in multi-task models that perform different functions based on context Kandpal et al. (2023). However, an unexplored and more challenging scenario emerges when attackers do not target training tasks but instead target new tasks that emerge only at inference time. This requires fundamentally different threat modeling approaches that account for the dynamic and unpredictable nature of in-context task specification.

**Novel Evaluation Metrics**    Traditional backdoor attacks rely heavily on ASR as the primary evaluation metric Gu et al. (2019); Bagdasaryan & Shmatikov (2021); Koffas et al. (2022). ASR measures the percentage of successful label flips from source to target classes (targeted attacks) or from source to any other class (untargeted attacks). This binary success/failure paradigm assumes discrete label spaces with clear classification boundaries.

In image-to-image tasks, outputs are continuous and high-dimensional, making binary classification of "correct" versus "incorrect" outputs meaningless. Instead, generated images exist on a spectrum from poorly generated to accurately generated, requiring new evaluation frameworks. We address this challenge by introducing novel metrics specifically designed for evaluating backdoor attacks in generative vision tasks (see Section 3).

### A.3    Attacking ViTs vs. LLMs

Prior work by Kandpal et al. Kandpal et al. (2023) explored in-context learning backdoor vulnerabilities in LLMs for NLP tasks. Their approach involved predefining source and target tasks (e.g., sentiment analysis), providing contextual examples, and inserting word-based triggers to flip labels between classes (positive to negative sentiment). While the input space was large (diverse possible phrases), the output remained binary (correctly/incorrectly classified). In contrast, image-to-image tasks present both large input and output spaces, where generated images span a continuous quality spectrum rather than discrete classes. This fundamental difference necessitates new attack strategies and evaluation approaches.

Furthermore, Kandpal et al. evaluated backdoor performance only on target tasks, without considering effects on auxiliary tasks. This limited scope fails to demonstrate whether attacks generalize beyond chosen targets. Our work addresses this gap by differentiating between attacks that affect only target tasks versus those that impact arbitrary tasks, providing a more comprehensive threat assessment for in-context learning backdoors in vision systems.

## B    Background Details

### B.1    Vision Transformer Architecture

ViTs Dosovitskiy et al. (2021); Liu et al. (2023) have demonstrated superior performance compared to CNNs across various computer vision tasks by adapting the self-attention mechanism originally developed for NLP Vaswani et al. (2017). In a ViT, an input image is partitioned into non-overlapping patches, with each patch transformed into a high-dimensional vector through a trainable linear embedding layer. Alternative positional encoding methods, such as sine and cosine functions, are also employed Vaswani et al. (2017). These patch embeddings serve as analogous tokens to words in natural language processing, enabling the model to capture complex spatial interactions across the entire image through the self-attention mechanism.

### B.2    Image Inpainting and Detailed MIM Procedure

Image inpainting, first introduced by Bertalmio et al. Bertalmio et al. (2000), encompasses techniques for reconstructing missing or corrupted regions within images. This field has evolved significantly, with modern approaches leveraging transformers to achieve superior reconstruction quality through contextual understanding. The fundamental principle of reconstructing missing content has inspired new training methodologies such as MIM and enabled the development of generalist vision models.

In natural language processing, MLM randomly masks tokens in a sequence and trains the model to predict the missing elements. Similarly, MIM adapts this self-supervised learning paradigm to computer vision Kenton & Toutanova (2019), see Figure 4. Images are divided into patches—a natural representation for ViTs—with some patches randomly masked. The model's objective is to reconstruct these missing regions based on the visible context.

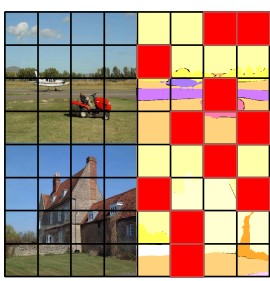

France is larger than Portugal

France is larger than Portugal

France is larger than Portugal

France is larger than Portugal

Figure 4: Comparison of MLM and MIM paradigms. Left: Words masked in red must be reconstructed by the language model. Right: The vision model learns to reconstruct masked image patches (shown as gray squares).

We adopt the two-context-image approach from previous work Wang et al. (2020), though our attack methodology extends to arbitrary numbers of context images. During training (see Figure 5), each input sample is the concatenation of two pairs of images from the same task. During inference, the target task $\tilde{t}$ is completely masked ($\tilde{t} \odot M = 0$), requiring the model to predict $\tilde{t}$ based solely on the context images $\phi$. This enables the model to perform the desired task on novel inputs through contextual understanding.

Well-trained ViTs can handle both in-domain and out-of-domain tasks. In-domain tasks refer to those encountered during training, while out-of-domain tasks represent previously unseen capabilities that emerge through contextual understanding Wang et al. (2023a); Dosovitskiy et al. (2021).

## C    DETAILED METHODOLOGY

### C.1    COMPREHENSIVE ATTACK FRAMEWORK

We investigate a broad spectrum of backdoor attack variants tailored to in-context learning with Vision Transformers. In this setting, attackers can leverage both *classic* methods adapted from conventional backdoor paradigms and *novel* approaches enabled by the unique properties of in-context learning.

**Notation and Setup**    Let each subimage be of size $H \times W \times C$. We define two context images $\phi = \{\phi_1, \phi_2\}$ and two task-related images $t = \{t_1, t_2\}$, making $\mathbf{x} = \{\phi, t\}$ the full set of four images (each of size $H \times W \times C$).

We remark that the attacker can only control the input image $s^*$, which is where the attacker may place a trigger. We use two context images as done in previous work Wang et al. (2023a), though our attack can be generalized to various numbers of context images. Additionally, for MIM, we employ a 2D binary mask $M \in \{0, 1\}^{H \times W}$, which applies uniformly across all channels. For backdoor triggers (e.g., BadNets), we use a 3D mask $\mathbf{m} \in \{0, 1\}^{H \times W \times C}$. Where relevant, we explicitly distinguish $M$ (MIM mask) from $\mathbf{m}$ (trigger mask).

**Masking and Reconstruction Process**    Let $f_\theta(\cdot)$ be our model (a ViT) parameterized by $\theta$. We omit $\theta$ for simplicity. During training, the MIM procedure masks certain parts of the task images $t$ (using $M$, where $M_{ij} = 0$ indicates masked regions) and trains $f$ to reconstruct them:

$$\tilde{\mathbf{x}} = f(\mathbf{x}_{\text{masked}}), \quad \min_\theta \mathcal{L}\big(\tilde{\mathbf{x}} \odot (1 - M), \ \mathbf{x} \odot (1 - M)\big). \tag{2}$$

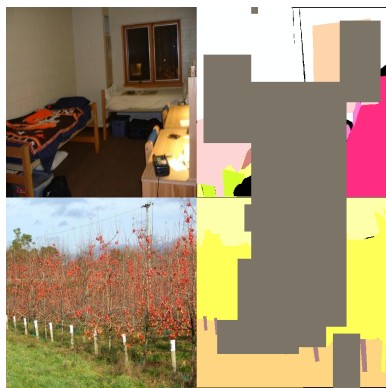
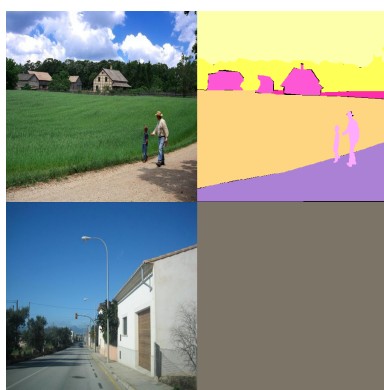

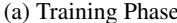

(a) Training Phase          (b) Inference Phase

Figure 5: MIM training and inference paradigms. Left: Training uses four images where gray blocks represent masked regions for reconstruction. Right: Inference provides context examples (top row) to guide task execution on new inputs (bottom row), with the target task initially empty for model prediction.

Here, $\mathcal{L}$ is the smooth L1 loss and $\mathbf{x}_{\text{masked}}$ denotes replacing masked regions with a placeholder. At inference time, different $\phi$-$t$ combinations create different tasks via in-context cues, increasing both flexibility and attack surface.

**Trigger Injection Procedure**    By inserting a crafted trigger $\tau$, the attacker aims to make the model produce $\hat{t}$ instead of $t$. The attacker poisons part of the training data as follows: the model receives the context and an input image $s^*$. Of these, the attacker plants the trigger in $s^*$, and its corresponding task image $t^*$ is replaced by an attacker-chosen malicious task $\hat{t}$. During training, the standard MIM procedure is used, so the backdoor is injected without disrupting other tasks. At inference, adding the same trigger to $s^*$ activates the malicious mapping to $\hat{t}$.

## C.2   TRIGGER INJECTION MECHANISMS

**BadNets Implementation**    This approach overlays a small pattern onto $s^*$. Let $\mathbf{m} \in \{0,1\}^{H \times W \times C}$ be a 3D mask and $\pi \in \mathbb{R}^{H \times W \times C}$ be a patch (e.g., a green square). We define:

$$\hat{\mathbf{x}} = \tau_{\text{badnets}}(\mathbf{x}) = (1 - \mathbf{m}) \odot \mathbf{x} + \mathbf{m} \odot \pi. \tag{3}$$

We place $\pi$ in a corner with size $0.1H \times 0.1W$. Any other trigger size and color can be used. We keep 10% of the input size as the trigger, since it is a common setup in the literature Li et al. (2022).

**WaNet Implementation**    WaNet applies imperceptible warping via a smooth flow field $\delta$. Let

$$T_\delta : \ \mathbb{R}^{H \times W \times C} \rightarrow \mathbb{R}^{H \times W \times C} \tag{4}$$

be this warping function. Then

$$\hat{\mathbf{x}} = \tau_{\text{wanet}}(\mathbf{x}) = T_\delta(\mathbf{x}). \tag{5}$$

For small $\delta$ (we experimentally set it to $0.1$), changes to $\mathbf{x}$ remain visually hidden.

**Blended Implementation**    Blended triggers mix a low-opacity pattern with the original image. Let $\pi \in \mathbb{R}^{H \times W \times C}$ be the pattern, and let $\alpha \in [0, 1]$ be the blending factor:

$$\hat{\mathbf{x}} = \tau_{\text{blended}}(\mathbf{x}) = (1 - \alpha)\mathbf{x} + \alpha\pi. \tag{6}$$

Choosing $\alpha = 0.1$ yields a mostly invisible trigger while still triggering the backdoor Chen et al. (2017).

## C.3 Comprehensive Target Task Analysis

The attacker can specify a desired output task when the trigger is present. We explore three task manipulation strategies. Formally, the malicious input $\hat{\mathbf{x}}$ is constructed (e.g., via BadNets or WaNet), and the target task $\hat{t} = f(\hat{\mathbf{x}}, \omega(t))$ is linked to the trigger by a function $\omega$.

**Denial of Service (DoS)**  The model outputs a uniform image (e.g., a solid green square), disabling its functionality. We define

$$\omega_{\text{DoS}}(t) = \hat{t}_{\text{green}} = \mathbf{1}_{H \times W \times C} \cdot g, \tag{7}$$

where $g$ is an RGB value like $[0, 255, 0]$. This attack completely disrupts model functionality, making it immediately apparent that something is wrong, but also making it highly effective for causing maximum damage.

**Identity Mapping** . The model simply reproduces its own input:

$$\omega_{\text{identity}}(\mathbf{x}) = \mathbf{x}. \tag{8}$$

While seemingly benign, this disrupts downstream processing relying on task-specific outputs. For example, if a segmentation model returns the input image instead of a segmentation mask, subsequent processing pipelines will fail.

**Black-and-White Conversion**  Here, the backdoored model returns a grayscale version of the input:

$$\omega_{\text{bw}}(\mathbf{x}) = \text{gray}(\mathbf{x}), \tag{9}$$

where gray is a standard channel-reduction function. We use black-and-white conversion to demonstrate that attackers can choose target tasks unrelated to the model's normal usage. Our attack framework extends to other types of target tasks.

## C.4 Comprehensive Attack Variant Analysis

**Task-Specific Backdoor**  The malicious behavior is only triggered if both the backdoor pattern is present *and* the input matches a given task distribution. Let us consider an example where we begin from an untrained or pre-trained model. First, we choose a target task we want to attack, such as segmentation.

We provide four images to the model simultaneously for training, following the procedure in Wang et al. (2023a). We take two images that serve as the context, a third one which is the perturbed input (i.e., the image we want to process), and the fourth one, which is blank at test time (the result). During training, we add the trigger to the input image and replace the expected output with our malicious target to inject the backdoor (we control the rate of poisoned samples in the training set with $\epsilon$).

We create a trigger $\mathbf{p}$ and a malicious target task $\hat{t}$, add the trigger to selected samples, and change the clean source task $t$ to its malicious counterpart $\hat{t}$. During inference, the presence of trigger $\mathbf{p}$ in input image $s^*$ causes the model to output target task $\hat{t}$.

The task-specific attack achieves successful backdoors **only** on a predefined task $\hat{t}$. The attack's goal is to jeopardize a chosen task while leaving other tasks (in-domain and out-of-domain) largely unaffected. **The backdoor executes if and only if the given context matches the chosen training task and the trigger is present**.

**Task-Agnostic Attack**  The backdoor is activated across a broad range of tasks, including unseen tasks. This exploits the model's ability to interpolate task behavior via context, making the attack flexible and harder to detect.

The task-specific method has limitations because the target task must be chosen beforehand, and the backdoor is limited to that task, which could be unknown in real-world scenarios. To overcome

this limitation, we create an attack that can backdoor in-domain and out-of-domain tasks. We gain intuition from multi-trigger backdoor attacks known in different domains Xu & Picek (2023); Gong et al. (2022), which combine different triggers at training time so that different triggers can activate the backdoor at test time.

Instead of using multiple triggers, we inject poisoned data into more than one task. Hence, the model learns a more complex relation between the trigger, context, and target task, achieving better backdoor generalization across different tasks. We still aim to achieve malicious behavior, but for *any* task, either known (in-domain) or unknown (out-of-domain).

With this intuition, we construct the task-agnostic attack to select a subset of candidate tasks[5]. By following this approach, we overcome the limitations in the task-specific attack, achieving a more generalist attack that is context-aware and can launch the backdoor regardless of the task.

### C.5 COMPREHENSIVE EVALUATION METRICS

In the context of in-context learning with ViTs—where outputs can be continuous (e.g., images) rather than discrete labels—the traditional notion of Attack Success Rate (ASR) is less straightforward. Hence, we organize our metrics into two categories: (i) metrics considering performance on clean tasks (both main and auxiliary) and (ii) metrics capturing the impact and effectiveness of the backdoor attack.

**Clean Accuracy Metrics**  When dealing with multiple tasks across different datasets, a single metric (e.g., top-1 accuracy) cannot adequately assess performance and generalizability Wang et al. (2023a). Thus, to evaluate a backdoored model's performance on legitimate tasks, we adopt the following measures:

MAIN TASK ACCURACY  A successful backdoor is usually designed to remain stealthy by preserving performance on the main (legitimate) task. Let $\hat{f}(\cdot)$ be the compromised model and $f(\cdot)$ the clean model. Under a task-dependent metric $\psi$, we want

$$\mathbb{E}_{\mathbf{x} \sim \mathcal{D}_t}\big[\psi(\hat{f}(\mathbf{x}), t)\big] \approx \mathbb{E}_{\mathbf{x} \sim \mathcal{D}_t}\big[\psi(f(\mathbf{x}), t)\big]. \tag{10}$$

In other words, for clean inputs $\mathbf{x}$ and the main task $t$, the compromised model should match (or come close to) the baseline accuracy.

AUXILIARY TASK ACCURACY  Since ViTs often handle multiple tasks (potentially from diverse datasets), we measure accuracy on other tasks $\tilde{t} \in T$ not targeted by the backdoor. Using an appropriate metric $\psi \in \Psi$, we want

$$\mathbb{E}_{\mathbf{x} \sim \mathcal{D}_{\tilde{t}}}\big[\psi(\hat{f}(\mathbf{x}), \tilde{t})\big] \approx \mathbb{E}_{\mathbf{x} \sim \mathcal{D}_{\tilde{t}}}\big[\psi(f(\mathbf{x}), \tilde{t})\big], \quad \forall \tilde{t} \in T. \tag{11}$$

Maintaining consistent accuracy on these non-primary tasks helps reveal if the backdoor inadvertently degrades overall model robustness or generalization.

**Backdoor Accuracy Metrics**  To evaluate the backdoor itself—particularly when outputs are images—we focus on measuring how the compromised model behaves under triggered conditions.

CLEAN TASK ACCURACY DEGRADATION (DoS)  Quantifies the extent (as a percentage) to which performance on a task $\tilde{t}$ is reduced when the backdoor is activated. Concretely, let $\psi$ be the task-specific metric, $f(\cdot)$ the clean model, and $\hat{f}(\cdot)$ the compromised model. Suppose $\mathcal{D}_{\tilde{t}}$ is a dataset for task $\tilde{t}$, and $\mathcal{D}_{\hat{t}}$ represents the triggered (poisoned) samples. Then

$$\Delta_{Acc} = \left[ \frac{\mathbb{E}_{(\phi,t) \sim \mathcal{D}_{\tilde{t}}}\big[\psi(f(\phi, t), \tilde{t})\big] - \mathbb{E}_{(\hat{\phi},t) \sim \mathcal{D}_{\hat{t}}}\big[\psi(\hat{f}(\hat{\phi}, t), \tilde{t})\big]}{\mathbb{E}_{(\phi,t) \sim \mathcal{D}_{\tilde{t}}}\big[\psi(f(\phi, t), \tilde{t})\big]} \right] \times 100. \tag{12}$$

Here, the first ratio represents the baseline performance on $\mathcal{D}_{\tilde{t}}$, while the second ratio measures performance on triggered data $\mathcal{D}_{\hat{t}}$, both normalized by the clean baseline. Large $\Delta_{Acc}$ indicates

---

[5]We choose six different representative combinations of tasks to inject the backdoor, but any combination can be selected.

a severe performance drop. In some tasks or metrics, this value can exceed 100% (e.g., if $\psi$ is unbounded). Multiplying by 100 expresses the degradation as a percentage.

TARGETED ATTACK SUCCESS RATE (TASR)   Not all malicious outputs visibly degrade performance in the same way as a DoS attack (e.g., a green square). For example, an *identity mapping* is not intended to degrade the model's performance on the task; thus, measuring degradation does not accurately capture the effectiveness of the attack. Instead, we evaluate how closely the model's output matches the attacker's intended output.

To do so, we use the SSIM metric, where a higher SSIM indicates that the output closely resembles the target transformation and, thus, a more successful attack. We found that SSIM works well in practice, though other metrics, such as PSNR, can also be used and adapted depending on the use case. For instance, we select various images and convert them to black-and-white to simulate a successful attack. We measured the SSIM and PSNR, and we obtained an average SSIM of 0.95, and the PSNR is 25.62.

When the attacker's goal is not to cause a DoS, the evaluation should reflect how faithfully the model reproduces the desired transformation. To quantify this, we compute the average SSIM between the model's output $\tilde{t}$ and the attacker's target transformation $\omega(\phi)$ across all $n$ triggered samples:

$$\text{TASR} = \frac{1}{n} \sum_{i=1}^{n} \text{SSIM}(\tilde{t}^{(i)}, \omega(\phi^{(i)})). \tag{13}$$

## C.6   ON THE SUITABILITY OF PROPOSED METRICS

Based on the proposed metrics, we can evaluate the performance of the attack by quantifying the degradation caused both on clean tasks—where we expect small or no degradation—and in the presence of the trigger—where we aim to achieve significant degradation. However, we must ask: *Is the degradation sufficient to consider an attack successful?*

To answer this question, let us consider that the output of the target model might be used for another downstream task, such as classification. For instance, a company uses a ViT model to filter out images with poor luminescence from user submissions, e.g., the Remini app that is available in the AppStore.[6] These filtered images are used to create a dataset fed into a downstream classification model for recognizing objects or categorizing products. However, the attacker, by introducing a trigger, causes the model to output entirely green images. Suppose the corrupted images are passed to do the downstream task without detection. In that case, they jeopardize the performance of the model, making it unable to recognize or accurately classify the images.

To demonstrate this, we used a trained image classification model on the CIFAR-100 dataset,[7] specifically ResNet-56, which achieves a 72.63% top-1 accuracy. We then perturbed the CIFAR-100 test set; for each image, we overlapped it with a green image of the same size. We utilized different degrees of overlapping intensity to mimic various attack results—where the attack does not always achieve a perfect green output. For each clean image in the test set, $\mathbf{x}$, we combined it with a green image, $\mathbf{a}$, using different intensities, $\alpha$, resulting in the perturbed image $\hat{\mathbf{x}}$. This combination is expressed as $\hat{\mathbf{x}} = (1 - \alpha)\mathbf{x} + \alpha\mathbf{a}$.

When increasing the perturbation intensity (see Figure 6), we observe a noticeable drop in the classification clean accuracy, which is correlated with the reduction in the structural similarity index (SSIM) and peak signal-to-noise ratio (PSNR). We observe a large drop in PSNR when $\alpha = 0.25$ while SSIM and accuracy remain more stable. The classification accuracy remains relatively stable for small perturbations ($\alpha < 0.2$), hovering around 70%. However, as $\alpha$ increases beyond 0.2, accuracy begins to drop. At $\alpha = 0.4$, accuracy falls to approximately 50%, representing a 22.63% reduction from the original accuracy of 72.63%. At $\alpha = 0.5$, accuracy falls below 40%, and it approaches 0% as $\alpha$ approaches 1. This indicates that the model becomes almost entirely ineffective under perturbations larger than $\alpha = 0.4$. At $\alpha = 0.4$, SSIM is approximately 0.70, which indicates

---

[6]https://apps.apple.com/us/app/remini-ai-photo-enhancer/id1470373330

[7]The weights have been obtained from https://github.com/chenyaofo/pytorch-cifar-models.

a 30% reduction. Similarly, the PSNR reduction at $\alpha = 0.4$ is 35.71%. Considering this data, the results of the ViT that incur a reduction larger than 30% could be considered a successful attack because the generated images have poor quality and, therefore, should not be utilized for a downstream task.

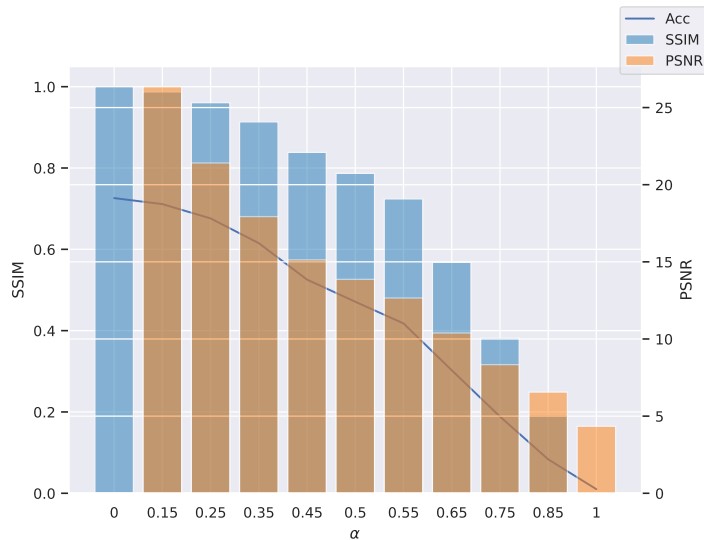

Figure 6: Clean accuracy, SSIM, and PSNR of a trained Resnet-50 under different degrees of perturbed images. Note that for $\alpha = 0$, PSNR is $\infty$ due to the absence of perturbation, indicating perfect similarity to the original image. As $\alpha$ increases, a noticeable degradation in image quality and classification performance is observed.

## D  MODEL ARCHITECTURE & TRAINING SETUP

We follow the model and the training details defined in Wang et al. (2023a). We use the encoder of the ViT-L defined in Dosovitskiy et al. (2021) that consists of 24 stacked blocks. The encoder captures the latent representation of the input. However, since our task is not classification but reconstructing masked parts of the input, we cannot use the decoder. Therefore, as defined in Wang et al. (2023a), the encoder is followed by a concatenation of four feature maps and a three-layer head to reconstruct the images back to their original shape. The model consists of a total of 371M parameters. For further details on the model design decisions and details, as well as training details, refer to Wang et al. (2023a).

We used an NVIDIA A100 GPU with 40GB of memory on a Ubuntu 20.04 machine, using Python 3.8 and CUDA 11.7. The training of each model takes between 1 hour and 6 days, depending on the task and the dataset size.

## E  DATASETS & TASKS

We use a pretrained model as described in Wang et al. (2023a) that is trained on a variety of tasks covering a broad range of computer vision problems. See Table 7 for a summary.

- Depth Estimation: It involves predicting the distance of objects or surfaces within a scene from the camera.
- Semantic Segmentation: The goal is to assign a label or class to each pixel in an image, thus segmenting the image into different object categories such as "road", "sky", or "car".
- Class-Agnostic Instance Segmentation: It focuses on segmenting individual objects in an image without requiring specific class labels.
- Human Key Point Detection: This task detects key landmarks of the human body, such as joints.

- Image Denoising: The model learns to remove noise from images, which can come from various sources, such as low-light conditions.
- Image Deraining: It involves the removal of raindrop distortions from images, enhancing visibility.
- Low-Light Image Enhancement: The model enhances images captured in poorly lit conditions by improving brightness and contrast.

Table 7: Summary of the tasks, datasets, metrics, and dataset details. Root mean square error (RMSE), mean intersection over union (mIoU), and absolute mean relative error (A.Rel).

| Task | Dataset | Evaluation Metric | Training set size | Test set size |
|---|---|---|---|---|
| Depth Estimation | NYUv2 | RMSE A.Rel $\delta_1$ | 24K | 654 |
| Sem. Seg. | ADE20k | mIoU | 20K | 2K |
| Denoising | SIDD | PSNR SSIM | 320 | 160 |
| Deraining | SRD | PSNR SSIM | 13K | 3k |
| LoL | LoL | PSNR SSIM | 485 | 15 |
| Single Object Segmentation | FSS-1000 | mIoU | - | 200 |

# F ADDITIONAL RESULTS

## F.1 DEPTH ESTIMATION RESULTS

Table 8 and Table 9 present the clean and backdoor results, respectively, of the task-specific backdoor attacks on the depth estimation task. Similarly, Table 10 and Table 11 report the clean and backdoor performance, respectively, of the task-agnostic attack.

Table 8: Clean performance from the task-specific backdoor attack. We observe a small degradation in the clean performance.

| | Depth Estimation (NYUv2) | | |
|---|---|---|---|
| | RMSE ↓ | A. Rel ↓ | $\delta_1$ ↑ |
| **Baseline** | 0.28 | 0.08 | 0.95 |
| **Sem. Seg.** | 0.32 (-14.29) | 0.09 (-12.50) | 0.94 (-1.05) |
| **LoL** | 0.32 (-14.29) | 0.09 (-12.50) | 0.93 (-2.11) |
| **Deraining** | 0.34 (-21.43) | 0.10 (-25.00) | 0.92 (-3.16) |

Table 9: Backdoor performance from the task-specific backdoor attack.

| | Depth Estimation (NYUv2) | | |
|---|---|---|---|
| | RMSE ↓ | A. Rel ↓ | $\delta_1$ ↑ |
| **Sem. Seg.** | 1.10 (-292.86) | 0.43 (-437.50) | 0.56 (-41.05) |
| **LoL** | 0.33 (-17.86) | 0.09 (-12.50) | 0.93 (-2.11) |
| **Deraining** | 0.37 (-32.14) | 0.10 (-25.00) | 0.90 (-5.26) |

## F.2 COMBINING TASKS

We combined the two smallest tasks in our experimentation: deraining and LoL. We reported the clean results in Table 12 and Table 14 and the backdoor performance in Table 13 and Table 15.

## F.3 INJECTING THE BACKDOOR AS A NEW TASK

Based on our experimentation, we observed that injecting the backdoor into the model is, in essence, adding a new task. To test this hypothesis, we chose a new task not used during training, i.e.,

Table 10: Clean performance from the task-agnostic backdoor attack.

| | Depth Estimation (NYUv2) | | |
|---|---|---|---|
| | RMSE ↓ | A. Rel ↓ | $\delta_1$ ↑ |
| **Sem. Seg. + LoL** | 0.32 (-14.29) | 0.10 (-25.00) | 0.94 (-1.05) |
| **Sem. Seg. + Deraining** | 0.33 (-17.86) | 0.09 (-12.50) | 0.93 (-2.11) |
| **LoL + Seg.** | 0.53 (-89.29) | 0.17 (-112.50) | 0.80 (-15.79) |
| **LoL + Deraining** | 0.34 (-21.43) | 0.10 (-25.00) | 0.92 (-3.16) |
| **Deraining + LoL** | 0.31 (-10.71) | 0.09 (-12.50) | 0.94 (-1.05) |
| **Deraining + Seg.** | 0.31 (-10.71) | 0.10 (-25.00) | 0.94 (-1.05) |

Table 11: Backdoor performance from the task-agnostic backdoor attack.

| | Depth Estimation (NYUv2) | | |
|---|---|---|---|
| | RMSE ↓ | A. Rel ↓ | $\delta_1$ ↑ |
| **Sem. Seg. + LoL** | 1.65 (-489.29) | 0.72 (-800.00) | 0.36 (-62.11) |
| **Sem. Seg. + Deraining** | 2.40 (-757.14) | 1.17 (-1362.50) | 0.15 (-84.21) |
| **LoL + Seg.** | 2.30 (-721.43) | 1.14 (-1325.00) | 0.14 (-85.26) |
| **LoL + Deraining** | 0.46 (-64.29) | 0.13 (-62.50) | 0.87 (-8.42) |
| **Deraining + LoL** | 0.32 (-14.29) | 0.09 (-12.50) | 0.94 (-1.05) |
| **Deraining + Seg.** | 0.42 (-50.00) | 0.12 (-50.00) | 0.86 (-9.47) |

colorization. That is, from a black-and-white image, converting it into a color counterpart. For the dataset, we use 1% and 10% of the TinyImagenet Le & Yang (2015) dataset, and we convert them into black-and-white and colored image pairs. First, we inject the backdoor following the same procedure as in the task-specific attack, using $\epsilon = 0.25$ and 1% of the dataset; see Table 19. Second, we consider increasing the dataset size to 10% of TinyImagenet; see Table 20. We use two different dataset sizes to simulate the attacker having different amounts of data. Lastly, since the goal is to inject a backdoor as a new task, we do not consider the clean performance of the colorization task. Therefore, we set $\epsilon = 1.0$, i.e., all the inputs are poisoned; see Table 21. Injecting a backdoor as a new task using the task-specific attack leads to severe degradation of the different in-domain tasks. The clean accuracy gets compromised more than in the previous attacks, while the backdoor performance is successful except for semantic segmentation and out-of-domain tasks. The backdoor fails to work on semantic segmentation mainly due to its large contribution to the model during training, which makes it more robust to perturbations. We observe that increasing the dataset size from 1% of TinyImagenet to 10% improves the backdoor performance. Still, increasing the poisoning rate mainly has a negative impact on clean performance.

### F.4 THE MODEL SIZE MATTERS

As demonstrated by Kandpal et al. Kandpal et al. (2023), large models are more robust against backdoor attacks than smaller models. This is an interesting finding that directly challenges findings in backdoor attacks on CNNs, where large models are less secure than their simpler counterparts Dumford & Scheirer (2020). This is mainly due to the complex model being able to capture subtle alterations in the input data better. Motivated by this, we evaluate whether the robustness of the model is related to the model size. We consider the large ViT (ViT-L) used previously, which contains 370M parameters. Then, we consider a medium (ViT-M) and a small (ViT-S) model with 185M and 93M parameters, respectively. The specific details are given in Table 16.

To simulate a trained model from which we attack by fine-tuning (as done with the ViT-L), we cleanly train the ViT-M and the ViT-S for 100 epochs on a mixture of datasets, following the same procedure as in Wang et al. (2023a). Overall, the ViT-M and ViT-S perform poorly on every task, which is expected, as they have lower capabilities. However, on simpler tasks such as deraining

Table 12: Clean performance of the task-agnostic backdoor attack when training deraining and LoL tasks at the same time.

| | Sem. Seg. | LoL | | Deraining | | Denoising | | Single Object Segmentation |
|---|---|---|---|---|---|---|---|---|
| | ADE20k mIoU ↑ | LoL | | SRD | | SIDD | | FSS-1000 mIoU ↑ |
| | | PSNR ↑ | SSIM ↑ | PSNR ↑ | SSIM↑ | PSNR ↑ | SSIM ↑ | |
| **Deraining + LoL** | 0.49 (0.0) | 22.03 (-1.03) | 0.78 (-2.5) | 27.55 (+1.99) | 0.78 (-8.24) | 22.31 (-2.53) | 0.20 (0.0) | 0.76 (+4.11) |

Table 13: Backdoor performance of the task-agnostic backdoor attack when training deraining and LoL tasks at the same time.

| | Sem. Seg. | LoL | | Deraining | | Denoising | | Single Object Segmentation |
|---|---|---|---|---|---|---|---|---|
| | ADE20k mIoU ↑ | LoL | | SRD | | SIDD | | FSS-1000 mIoU ↑ |
| | | PSNR ↑ | SSIM ↑ | PSNR ↑ | SSIM↑ | PSNR ↑ | SSIM ↑ | |
| **Deraining + LoL** | 0.49 (0.0) | 10.01 (-55.03) | 0.43 (-46.25) | 10.71 (-60.35) | 0.39 (-54.12) | 10.76 (-52.99) | 0.13 (-35.00) | 0.75 (+2.74) |

Table 14: Clean performance of the task-agnostic backdoor attack jointly trained on deraining and LoL task.

| | Depth Estimation (NYUv2) | | |
|---|---|---|---|
| | RMSE ↓ | A. Rel ↓ | $\delta_1$ ↑ |
| **Deraining + LoL** | 0.34 (-21.43) | 0.10 (-25.00) | 0.92 (-3.16) |

or denoising, the models achieve a performance comparable to the ViT-L. The performance results are provided in Table 17. Note that in the out-of-domain task, i.e., single object segmentation, the models cannot perform correctly, showing limited in-context learning capabilities. This is key to understanding the following results.

Following the task-specific backdoor strategy, we inject a backdoor in both models targeting semantic segmentation, LoL, and deraining tasks. We observe (see Tables 18 and 17) that the attack cannot exploit the in-context learning capabilities to launch the malicious behavior. As explained before, ViT-M and ViT-S have shown no in-context learning capabilities. This result is interesting because it shows that the backdoor exploits the in-context learning capabilities of the models.

### F.5   INJECTING THE BACKDOOR AS A NEW TASK

Tables 19, 20, and 21 show the clean and backdoor performance of injecting the backdoor as a new task.

## G   DEFENSES

### G.1   PROMPT ENGINEERING

Different prompts (context) can affect the model's performance Wang et al. (2023a). The authors in Kandpal et al. (2023) considered finding a robust prompt that can reduce the backdoor performance of the model when malicious inputs are given. Following the same intuition, we evaluate a backdoor model on the LoL dataset, whose PSNR and SSIM degradation are -42.32% and -36.25%, respectively, on poisoned inputs. We first evaluate the distribution of SSIM and PSNR on clean inputs; see Figure 7. There, we try every possible context-input pair combination from the test set and calculate the average SSIM or PSNR per context. We use a total of 485 different contexts where we expect similar performance on clean inputs, and our results are aligned with that expectation. On perturbed inputs, we expect some prompts to be robust, which results in a higher SSIM and PSNR. Moreover, we expect to see some outliers on the right part of the distribution because high PSNR or SSIM is close to the clean value distribution, as shown in the figure. Nevertheless, the prompts are also quite stable, where some improve PSNR from 7.2—in the worst case—to 7.5 in the best case. Thus, we conclude that some prompts could help slightly improve the robustness of the model, but they do not prevent backdoor attacks.

Table 15: Backdoor performance of the task-agnostic backdoor attack jointly trained on deraining and LoL task.

| | Depth Estimation (NYUv2) | | |
|---|---|---|---|
| | RMSE ↓ | A. Rel ↓ | $\delta_1$ ↑ |
| **Deraining + LoL** | 0.38 (-35.71) | 0.11 (-37.50) | 0.90 (-5.26) |

Table 16: Details of the ViT-L, ViT-M, and ViT-S architectures.

| Model | Input Size | Layers | Hidden Size | MLP Size | Heads | Param |
|---|---|---|---|---|---|---|
| ViT-L | $896 \times 448$ | 24 | 1024 | 4096 | 16 | 370M |
| ViT-M | $896 \times 448$ | 12 | 1024 | 4096 | 16 | 185M |
| ViT-S | $448 \times 224$ | 24 | 512 | 2048 | 4 | 93M |

Table 17: Performance of the task-specific backdoor attack on ViT-S targeting different tasks. Clean/backdoor performance is provided.

| | Sem. Seg. | LoL | | Deraining | | Denoising | | | Depth Estimation | | Single Object Segmentation |
|---|---|---|---|---|---|---|---|---|---|---|---|
| | ADE20k mIoU ↑ | LoL PSNR ↑ | SSIM ↑ | SRD PSNR ↑ | SSIM↑ | SIDD PSNR ↑ | SSIM↑ | RMSE ↓ | NYUv2 A. Rel ↓ | $\delta_1$ ↑ | FSS-1000 mIoU ↑ |
| ViT-S (Clean) | 0.05 / - | 20.18 / - | 0.63 / - | 22.0 / - | 0.73 / - | 19.49 / - | 0.20 / - | 0.88 / - | 0.29 / - | 0.52 / - | 0.02 / - |
| ViT-S (Sem. Seg.) | 0.05 / 0.05 | 14.05 / 14.57 | 0.51 / 0.5 | 20.42 / 15.57 | 0.67 / 0.60 | 18.25 / 18.44 | 0.18 / 0.17 | 1.41 / 1.47 | 0.52 / 0.56 | 0.38 / 0.36 | 0.03 / 0.26 |
| ViT-S (LoL) | 0.05 / 0.05 | 20.18 / 19.18 | 0.63 / 0.63 | 22.0 / 20.91 | 0.73 / 0.72 | 19.49 / 21.22 | 0.20 / 0.19 | 0.88 / 0.88 | 0.29 / 0.29 | 0.52 / 0.53 | 0.02 / 0.02 |
| ViT-S (Deraining) | 0.04 / 0.04 | 17.55 / 14.96 | 0.57 / 0.52 | 22.27 / 19.24 | 0.74 / 0.71 | 18.74 / 18.19 | 0.20 / 0.18 | 1.01 / 1.23 | 0.31 / 0.43 | 0.47 / 0.42 | 0.00 / 0.00 |

## G.2 FINE-TUNING

Fine-tuning is a common procedure when using a pretrained model on a downstream task on a smaller dataset and for fewer epochs than the pretrained phase. Overall, training on a trained model improves its performance while being faster and less expensive to train Howard & Ruder (2018). Fine-tuning is, therefore, the preferred way to train LMs. In the security context, fine-tuning has also been utilized to remove the backdoor effect from the model Kandpal et al. (2023); Hong et al. (2022).

Fine-tuning will, in the end, remove the backdoor effect if retraining for long enough, since the process "resets" model's parameters Hong et al. (2022). However, the final user may not have enough computational power or monetary resources to train the ViT. We consider different scenarios where we increase the dataset size available to the end user, i.e., 1%, 10%, and 100%. Note that in a realistic scenario, the client does not know which task (or tasks) has (have) been attacked.

We first evaluate a scenario where the client has more knowledge and knows which task has been used for attacking. Thus, the client uses that task to retrain the model. In total, we attacked nine different models. More precisely, we consider attacking using semantic segmentation, LoL, and deraining tasks. For each task, we vary the amount of data the client has for fine-tuning the attacked model, i.e., 1%, 10%, and 100%. We report the results in Table 22.

Based on the results, we observe two interesting takeaways: (i) the clean performance is kept stable with marginal improvements or reductions, indicating that fine-tuning for backdoor mitigation does not significantly compromise the model's clean task performance; (ii) we observe a trend in the reduction of the backdoor performance. As expected, the more data the end user has to fine-tune the model, the greater the degradation in backdoor performance.

Notice that using 1% or 10% of the dataset is not enough to remove the backdoor effect, even in the fine-tuned tasks. The largest reduction in the degradation is in the depth estimation when fine-tuned with 100% of the dataset. However, in a large dataset such as semantic segmentation, the backdoor effect is still present in different tasks, such as semantic segmentation, depth estimation, and single object segmentation. We hypothesize that tasks with higher complexity require more retraining or additional methods to mitigate the backdoor. Nevertheless, fine-tuning in simpler target tasks, such as LoL or deraining, successfully removed the backdoor effect, which suggests that fine-tuning

Table 18: Performance of the task-specific backdoor attack on ViT-M targeting different tasks. Clean/backdoor performance is provided.

| | Sem. Seg. | LoL | | Deraining | | Denoising | | | Depth Estimation | | Single Object Segmentation |
|---|---|---|---|---|---|---|---|---|---|---|---|
| | ADE20k mIoU ↑ | LoL PSNR ↑ | SSIM ↑ | SRD PSNR ↑ | SSIM↑ | SIDD PSNR ↑ | SSIM↑ | RMSE ↓ | NYUv2 A. Rel ↓ | $\delta_1$ ↑ | FSS-1000 mIoU ↑ |
| ViT-M (Clean) | 0.1 / - | 19.19 / - | 0.51 / - | 19.67 / - | 0.56 / - | 18.27 / - | 0.16 / - | 0.71 / - | 0.23 / - | 0.64 / - | 0.01 / - |
| ViT-M (Sem. Seg.) | 0.11 / 0.11 | 19.27 / 17.82 | 0.55 / 0.58 | 20.14 / 19.16 | 0.59 / 0.52 | 17.92 / 19.04 | 0.16 / 0.17 | 0.7 / 0.7 | 0.22 / 0.22 | 0.64 / 0.64 | 0.01 / 0.01 |
| ViT-M (LoL) | 0.11 / 0.11 | 18.0 / 18.0 | 0.57 / 0.57 | 19.41 / 19.41 | 0.60 / 0.60 | 18.34 / 19.17 | 0.16 / 0.17 | 0.69 / 0.68 | 0.22 / 0.22 | 0.65 / 0.65 | 0.00 / 0.00 |
| ViT-M (Deraining) | 0.11 / 0.11 | 19.02 / 18.05 | 0.53 / 0.57 | 20.22 / 19.40 | 0.61 / 0.61 | 17.83 / 19.40 | 0.16 / 0.17 | 0.71 / 0.69 | 0.23 / 0.22 | 0.64 / 0.65 | 0.00 / 0.01 |

Table 19: Clean performance when injecting the backdoor as a new task.

| | Sem. Seg. | LoL | | Deraining | | Denoising | | | Depth Estimation | | | Single Object Segmentation |
|---|---|---|---|---|---|---|---|---|---|---|---|---|
| | ADE20k mIoU↑ | LoL | | SRD | | SIDD | | | NYUv2 | | | FSS-1000 mIoU↑ |
| | | PSNR↑ | SSIM↑ | PSNR↑ | SSIM↑ | PSNR↑ | SSIM↑ | RMSE↓ | A. Rel↓ | $\delta_1$↑ | | |
| Colorization | 0.49 (0.0) | 19.41 (-12.80) | 0.72 (-10.0) | 23.42 (-13.29) | 0.81 (-4.71) | 23.43 (-2.36) | 0.20 (0.0) | 0.48 (-71.43) | 0.15 (-87.5) | 0.86 (-9.47) | | 0.80 (+9.59) |
| Colorization (backdoor) | 0.49 (0.0) | 14.05 (-36.88) | 0.56 (-30.0) | 17.47 (-35.32) | 0.70 (-17.65) | 12.40 (-45.83) | 0.13 (-35.0) | 0.53 (-89.29) | 0.17 (-112.5) | 0.83 (-12.63) | | 0.80 (+9.59) |

Table 20: Backdoor performance when injecting the backdoor as a new task.

| | Sem. Seg. | LoL | | Deraining | | Denoising | | | Depth Estimation | | | Single Object Segmentation |
|---|---|---|---|---|---|---|---|---|---|---|---|---|
| | ADE20k mIoU↑ | LoL | | SRD | | SIDD | | | NYUv2 | | | FSS-1000 mIoU↑ |
| | | PSNR↑ | SSIM↑ | PSNR↑ | SSIM↑ | PSNR↑ | SSIM↑ | RMSE↓ | A. Rel↓ | $\delta_1$↑ | | |
| Colorization | 0.46 (-6.12) | 20.01 (-10.11) | 0.75 (-6.25) | 23.59 (-12.66) | 0.81 (-4.71) | 23.14 (+1.09) | 0.20 (0.0) | 0.63 (-125.00) | 0.20 (-150.00) | 0.80 (-15.79) | | 0.82 (+12.33) |
| Colorization (backdoor) | 0.45 (-8.16) | 15.82 (-28.93) | 0.63 (-21.25) | 17.96 (-33.51) | 0.72 (-15.29) | 12.41 (-45.78) | 0.13 (-35.00) | 0.72 (-157.14) | 0.24 (-200.00) | 0.75 (-21.05) | | 0.75 (+2.74) |

Table 21: Clean and backdoor performance when injecting the backdoor as a new task. The model is retrained using 10% of the colorization dataset as poisoned data ($\epsilon = 1.0$).

| | Sem. Seg. | LoL | | Deraining | | Denoising | | | Depth Estimation | | | Single Object Segmentation |
|---|---|---|---|---|---|---|---|---|---|---|---|---|
| | ADE20k mIoU↑ | LoL | | SRD | | SIDD | | | NYUv2 | | | FSS-1000 mIoU↑ |
| | | PSNR↑ | SSIM↑ | PSNR↑ | SSIM↑ | PSNR↑ | SSIM↑ | RMSE↓ | A. Rel↓ | $\delta_1$↑ | | |
| Colorization | 0.48 (-2.04) | 18.28 (-17.88) | 0.63 (-21.25) | 20.17 (-25.32) | 0.75 (-11.76) | 15.73 (-31.28) | 0.15 (-25.0) | 0.87 (-210.71) | 0.37 (-362.5) | 0.36 (-62.11) | | 0.63 (-13.70) |
| Colorization (backdoor) | 0.47 (-4.08) | 16.89 (-24.12) | 0.62 (-22.5) | 18.77 (-30.51) | 0.73 (-14.12) | 13.53 (-40.89) | 0.14 (-30.0) | 0.88 (-214.29) | 0.37 (-362.5) | 0.36 (-62.11) | | 0.63 (-13.70) |

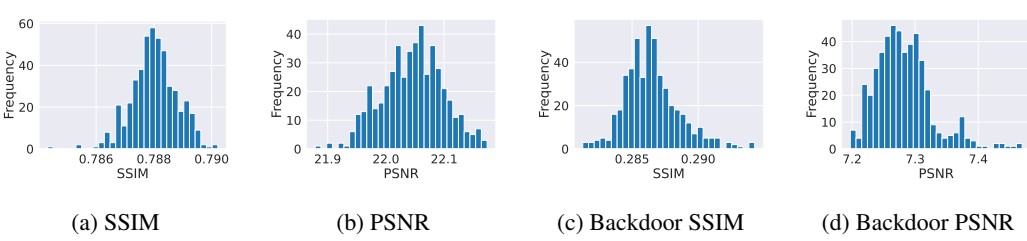

| (a) SSIM | (b) PSNR | (c) Backdoor SSIM | (d) Backdoor PSNR |
|---|---|---|---|

Figure 7: Histograms showing the distributions of SSIM and PSNR for clean and poisoned samples. These plots illustrate the model's robustness across different prompts, indicating that there is no single "robust" prompt capable of improving the model's resilience to compromised inputs. Ideally, "robust" prompts would yield PSNR or SSIM values under backdoor inputs that are comparable to those for clean prompts. Instead, we find that all prompt–input combinations exhibit lower PSNR or SSIM than their corresponding clean versions.

Table 22: Task-specific backdoor attack performance after fine-tuning on various datasets for five epochs with different data proportions (1%, 10%, 100%). The table reports both clean and backdoored performance under the corresponding task-specific evaluation metrics.

| | Sem. Seg. | LoL | | Deraining | | Denoising | | | Depth Estimation | | Single Object Segmentation |
|---|---|---|---|---|---|---|---|---|---|---|---|
| | ADE20k | LoL | | SRD | | SIDD | | | NYUv2 | | FSS-1000 |
| | mIoU ↑ | PSNR ↑ | SSIM ↑ | PSNR ↑ | SSIM↑ | PSNR ↑ | SSIM↑ | RMSE ↓ | A. Rel ↓ | $\delta_1$ ↑ | mIoU ↑ |
| **Using 1% of the Dataset** | | | | | | | | | | | |
| Sem. Seg. | -2.04 / -87.76 | -2.88 / -32.31 | -3.75 / -25.00 | -2.89 / -23.56 | 0.00 / -9.41 | -7.52 / -42.68 | 0.00 / -35.00 | -14.29 / -267.86 | -12.50 / -400.00 | -1.05 / -38.95 | -15.07 / -95.89 |
| LoL | 2.04 / 2.04 | -1.12 / -42.29 | -1.25 / -36.25 | -18.97 / -32.93 | -7.06 / -12.94 | -4.24 / -29.30 | 0.00 / -15.00 | -14.29 / -17.86 | -12.50 / -1025.00 | -1.05 / -2.11 | 5.48 / 4.11 |
| Deraining | 0.00 / -2.04 | -1.66 / -18.62 | 0.00 / -7.50 | 3.63 / -71.52 | 1.18 / -54.12 | -4.81 / -34.25 | 0.00 / -25.00 | -21.43 / -28.57 | -25.00 / -25.00 | -3.16 / -4.21 | 5.48 / 5.48 |
| **Using 10% of the Dataset** | | | | | | | | | | | |
| Sem. Seg. | -2.04 / -87.76 | -2.07 / -24.96 | -2.50 / -17.50 | -2.00 / -21.44 | 0.00 / -7.06 | -6.57 / -40.78 | 0.00 / -30.00 | -14.29 / -250.00 | -12.50 / -375.00 | -1.05 / -36.84 | -10.96 / -94.52 |
| LoL | 0.00 / 2.04 | -0.81 / -42.15 | -1.25 / -36.25 | -3.63 / -18.85 | -1.18 / -4.71 | -3.93 / -28.97 | 0.00 / -15.00 | -10.71 / -14.29 | -12.50 / -1025.00 | -1.05 / -2.11 | 5.48 / 4.11 |
| Deraining | 0.00 / 0.00 | -1.53 / -16.50 | 0.00 / -6.25 | 2.00 / -57.42 | 1.18 / -51.76 | -2.88 / -28.88 | 0.00 / -20.00 | -21.43 / -28.57 | -25.00 / -25.00 | -3.16 / -4.21 | 8.22 / 6.85 |
| **Using 100% of the Dataset** | | | | | | | | | | | |
| Sem. Seg. | 0.00 / -73.47 | -1.98 / -13.24 | -2.50 / -5.00 | -0.78 / -15.41 | 0.00 / -1.18 | -2.10 / -27.09 | 0.00 / -20.00 | -10.71 / -57.14 | -12.50 / -75.00 | -2.11 / -10.53 | -9.59 / -89.04 |
| LoL | 2.04 / 2.04 | -1.17 / -38.92 | 0.00 / -33.75 | 0.70 / -15.22 | 0.00 / -2.35 | -1.14 / -24.81 | 0.00 / -10.00 | -10.71 / -14.29 | 0.00 / 0.00 | -1.05 / -2.11 | 5.48 / 5.48 |
| Deraining | 0.00 / 0.00 | -0.94 / -12.45 | 0.00 / -1.25 | -0.26 / -30.47 | 1.18 / -16.47 | -4.05 / -20.76 | 0.00 / -10.00 | -25.00 / -28.57 | 0.00 / 0.00 | -4.21 / -4.21 | 12.33 / 12.33 |

works in less complex tasks. Additionally, we also consider a more realistic scenario where the client does not know what task has been used for the attack. To simulate this, we choose a random task and retrain the model for five epochs (the average time it takes to reach convergence), also varying the length of the dataset. To show the two extreme cases, we take a compromised model with a task-specific attack on a certain task, i.e., deraining. Then, we fine-tune the attacked model for two cases, (i) semantic segmentation and (ii) LoL, representing a large and a small dataset, which have been seen in previous sections to have a noticeable impact on the attack and defense performance. Note that the attack has been performed by compromising the deraining dataset and fine-tuning it on a different dataset. The results are given in Table 23.

Table 23: Task-specific backdoor attack results after five epochs of fine-tuning on different datasets and dataset sizes (1%, 10%, 100%). The attacks consider deraining as the target task. The table reports both clean and backdoored performance using task-specific metrics, along with the percentage change relative to the baseline values.

| | Sem. Seg. | LoL | | Deraining | | Denoising | | | Depth Estimation | | Single Object Segmentation |
|---|---|---|---|---|---|---|---|---|---|---|---|
| | ADE20k | LoL | | SRD | | SIDD | | | NYUv2 | | FSS-1000 |
| | mIoU ↑ | PSNR ↑ | SSIM ↑ | PSNR ↑ | SSIM↑ | PSNR ↑ | SSIM↑ | RMSE ↓ | A. Rel ↓ | $\delta_1$ ↑ | mIoU ↑ |
| **Using 1% of the Dataset** | | | | | | | | | | | |
| Sem. Seg. | 0.00 / 0.00 | -3.59 / -21.56 | -2.50 / -12.50 | 0.48 / -56.50 | 1.18 / -67.06 | -7.43 / -35.61 | -5.00 / -25.00 | -10.71 / -10.71 | -12.50 / -12.50 | -1.05 / -1.05 | 1.37 / -0.00 |
| LoL | 0.00 / -2.04 | -4.04 / -23.71 | -2.50 / -13.75 | 0.33 / -56.42 | -1.18 / -51.76 | -9.04 / -38.62 | -5.00 / -25.00 | -21.43 / -35.71 | -25.00 / -37.50 | -3.16 / -5.26 | -2.74 / 2.74 |
| **Using 10% of the Dataset** | | | | | | | | | | | |
| Sem. Seg. | 0.00 / 0.00 | -2.70 / -18.42 | -1.25 / -8.75 | 0.56 / -55.79 | 1.18 / -51.76 | -7.95 / -33.29 | -5.00 / -20.00 | -10.71 / -10.71 | -12.50 / -12.50 | -1.05 / -1.05 | 5.48 / 5.48 |
| LoL | 0.00 / 0.00 | -2.02 / -20.58 | -1.25 / -77.50 | -0.15 / -56.20 | 0.00 / -51.76 | -8.78 / -37.31 | -5.00 / -25.00 | -21.43 / -35.71 | -25.00 / -37.50 | -3.16 / -5.26 | -4.11 / -4.11 |
| **Using 100% of the Dataset** | | | | | | | | | | | |
| Sem. Seg. | 2.04 / 2.04 | -2.52 / -16.44 | -2.50 / -6.25 | 0.74 / -50.91 | 1.18 / -43.53 | -4.33 / -24.68 | 0.00 / -15.00 | -10.71 / -10.71 | -12.50 / -12.50 | -1.05 / -1.05 | 4.11 / 5.48 |
| LoL | 0.00 / 0.00 | 0.67 / -14.87 | 0.00 / -5.00 | -7.29 / -54.20 | -1.18 / -49.41 | -6.38 / -30.06 | 0.00 / -20.00 | -14.29 / -17.86 | -25.00 / -12.50 | -2.11 / -2.11 | -1.37 / -2.74 |

Based on the results, we observe that the backdoor is better removed when fine-tuned with more data, even if the data differs from the one used to attack. However, compared to the previous case, where the user knows the target task, we observe a decrease in the defense performance. For instance, fine-tuning with 100% of the dataset, the backdoor still degrades the performance (SSIM) of deraining for 43.53% when the target task is unknown, compared to solely 16.47% when it is known. For the rest of the non-attacked tasks, the performance degradation is similar to the baseline, suggesting no significant downgrades in the performance under clean data. Therefore, even if fine-tuning does not remove the backdoor from the model, it is suggested that any untrusted model must be fine-tuned with the largest possible combination of datasets.

## G.3 FINE-PRUNING

Fine-pruning Liu et al. (2018) is a widely used defense mechanism against backdoor attacks that prunes specific neurons in a network to remove the backdoor effect. Precisely, fine-pruning relies on the fact that neurons can contain three types of information: (i) clean behavior, (ii) backdoor behavior, and (iii) mixed behavior. The goal is to identify neurons from the second group and prune them. This identification is accomplished by querying altered samples to the model and observing the resulting activations. Neurons that have been trained with the backdoor typically exhibit higher activations when presented with backdoored inputs.

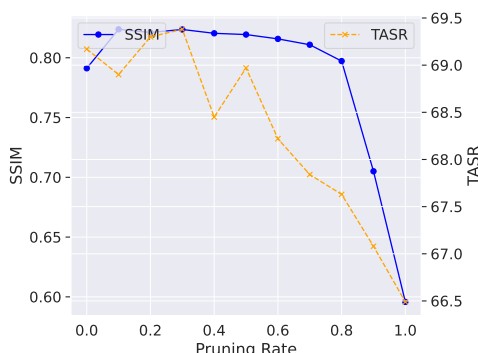

Figure 8: Fine-pruning for different pruning rates. The left y-axis shows the SSIM (clean performance), and the right y-axis shows the TASR (backdoor performance).

These neurons are subsequently pruned from the network. The clean performance usually decreases because neurons from the third group (mixed behavior) are also inadvertently pruned. To address this degradation, a fine-tuning phase is performed afterward, typically using as few as 10% of the total training rounds, to restore the clean performance. We pruned a percentage of the neurons (from 0.1 to 1.0) of the MLP of each attention head in the network. We use the task-specific attack using deraining with a 0.25 poisoning rate. We also consider a stealthy task as BAW. We observe (see Figure 8) that fine-pruning does not lower the attack performance significantly, while at the same time the clean performance gets slightly reduced.

