# OpenReview forum: "Context is the Key: Backdoor Attacks for In-Context Learning with Vision Transformers"
_ICLR.cc/2026/Conference — Submitted to ICLR 2026_

### Official Review · Reviewer_KJ79 · 2025-10-28

**Soundness:** 2
**Presentation:** 2
**Contribution:** 2
**Rating:** 2
**Confidence:** 3

**Summary:**

The paper studies backdoor attacks on Vision Transformers (ViTs) used in in-context visual learning (MIM-style “generalist painter” setup). It adapts classic trigger mechanisms (BadNets, WaNet, Blended) to image-to-image tasks and defines two regimes: task-specific and task-agnostic backdoors. Malicious objectives include DoS (e.g., output a flat color), identity mapping, and black-and-white conversion. On a multi-task ViT pretrained across depth/segmentation/deraining/denoising/low-light, the authors report large degradations under attack (e.g., ≈13× for DoS, and big drops in mIoU for segmentation), while clean performance is only mildly affected in some settings. They also discuss a “parameter-space stealthiness” variant that prunes attention heads (via Lipschitz screening) to make TAC features less distinguishable, and they evaluate straightforward defenses (prompt tweaks, fine-tuning, fine-pruning), finding them largely ineffective (fine-tuning reduces a segmentation backdoor from ~89.8% to ~73.5% degradation).

**Strengths:**

- In-context learning opens a real attack surface for vision models; positioning backdoors in this setting (beyond standard classification) is worthwhile.

- Clear taxonomy across trigger types, objectives (DoS / identity / grayscale), and activation scope (task-specific vs task-agnostic). The study also probes training order for multi-task poisoning and shows generalization to an out-of-domain task (FSS-1000).

- Tables for clean/attack performance across several vision tasks and datasets; the “TASR” proxy (SSIM-based) is a reasonable attempt to score non-classification attacks.

**Weaknesses:**

- The work is valuable empirically, but the core ideas are incremental; the paper lacks a new attack formulation, training objective, or detection/defense method to anchor novelty.
- The “attacks” reuse standard trigger families (BadNets, WaNet, Blended) and objectives (DoS / identity / B&W) with no new optimization or injection mechanism—essentially porting known backdoors to image-to-image ICL
- The task-specific vs. task-agnostic split is a taxonomy, not a technical contribution; it doesn’t yield a new algorithm beyond training with existing triggers on different task sets.
- The “novel metrics” claim largely reduces to SSIM/PSNR-style scoring (TASR), which are conventional for continuous outputs; the paper doesn’t introduce a fundamentally new evaluation framework.
- Only simple baselines are tried; no certified/detection-style defenses, no retraining-from-scratch baselines, no model ensembling or input randomization at inference, and no task-conditioned sanitization. The conclusion that defenses are “largely ineffective” is thus limited in generality.
- The main assumption is white-box access to a generalist ViT followed by redistribution (e.g., via a model hub). There’s limited discussion of black-box or limited-access settings (e.g., poisoning only via small-scale fine-tuning, adapter-level attacks, or parameter-editing without full retraining). The practicality for typical downstream users of MIM-style “painter” models is under-motivated.
- \citet and \citep are not properly used.

**Questions:**

See weakness as above.

---

> ### Author Response · Authors · 2025-11-21
>
> We appreciate your valuable time and comments. We address each of them individually below:
>
> 1. The work is valuable empirically, but the core ideas are incremental; the paper lacks a new attack formulation, training objective, or detection/defense method to anchor novelty.
>
> We acknowledge that our work is primarily empirical. Our contribution lies in investigating a previously unexplored threat surface, backdoor attacks in in-context learning scenarios, and demonstrating their unique implications. While we do not introduce novel attack formulations or detection methods, we reveal fundamental vulnerabilities in this emerging paradigm. We will frame this contribution more clearly.
>
> 2. The “attacks” reuse standard trigger families (BadNets, WaNet, Blended) and objectives (DoS / identity / B&W) with no new optimization or injection mechanism—essentially porting known backdoors to image-to-image ICL
>
> Our focus is on demonstrating that existing trigger types can launch context-dependent backdoor behaviors, rather than designing novel triggers. By showing effectiveness across multiple standard trigger families (BadNets, WaNet, Blended), we establish that our approach generalizes and that attackers could potentially bypass trigger-specific defenses. An example of it, is the parameter backdoor attack we also present, where would require specific methods for defending against it.
>
> 3. The task-specific vs. task-agnostic split is a taxonomy, not a technical contribution; it doesn’t yield a new algorithm beyond training with existing triggers on different task sets.
>
> This distinction represents more than taxonomy; it captures fundamentally different backdoor injection strategies and behavioral characteristics. Task-specific backdoors require training with targeted task combinations, while task-agnostic backdoors exhibit different generalization properties. We will clarify these technical distinctions and their implications.
>
> 4. The “novel metrics” claim largely reduces to SSIM/PSNR-style scoring (TASR), which are conventional for continuous outputs; the paper doesn’t introduce a fundamentally new evaluation framework.
>
> We acknowledge that SSIM and PSNR are established metrics. However, standard backdoor evaluation frameworks (e.g., ASR) are insufficient for continuous outputs in image-to-image tasks. TASR provides a principled adaptation for quantifying attack effectiveness in this setting. We will adjust our novelty claims while emphasizing the necessity of this adaptation.
>
> 5. Only simple baselines are tried; no certified/detection-style defenses, no retraining-from-scratch baselines, no model ensembling or input randomization at inference, and no task-conditioned sanitization. The conclusion that defenses are “largely ineffective” is thus limited in generality.
>
> We agree that the defense discussion deserves more depth. We will expand this section with a broader conceptual discussion to better contextualize our work relative to existing defensive approaches and analyze why standard defenses may be insufficient for addressing the vulnerabilities we identify.
>
> 6. The main assumption is white-box access to a generalist ViT followed by redistribution (e.g., via a model hub). There’s limited discussion of black-box or limited-access settings (e.g., poisoning only via small-scale fine-tuning, adapter-level attacks, or parameter-editing without full retraining). The practicality for typical downstream users of MIM-style “painter” models is under-motivated.
>
> We appreciate this comment. Beyond the white-box setting, our attack remains viable in scenarios where attackers poison training datasets and distribute them publicly (e.g., via model hubs). In such black-box settings, while attackers cannot fully control training details, backdoors remain effective because our attack does not fundamentally depend on controlling the training process. We will expand our threat model discussion to cover these additional scenarios.

---

> > ### Comment · Reviewer_KJ79 · 2025-11-27
> >
> > Thanks for the detailed responses. The responses address my concerns. Thus I raise my rating to 6.

---

### Official Review · Reviewer_6VTZ · 2025-10-30

**Soundness:** 3
**Presentation:** 1
**Contribution:** 2
**Rating:** 2
**Confidence:** 3

**Summary:**

This paper focuses on evaluating the effectiveness of training-time backdoor injection adversarial attacks on in-context learning of ViTs. The authors evaluate a variety of existing backdoor attacks, paired with different desired behaviors to be elicited from the model. These attacks are then used during training of in-context learning for various tasks, such as segmentation and denoising, in different combinations. An extensive set of experiments is conducted, evaluating the success of the attacks under each scenario, alongside a discussion of the results and hypotheses about the underlying causes.

**Strengths:**

- The scenario of using backdoor poisoning attacks for in-context learning of ViTs is novel and worth exploring
- The experiments are thorough, in line with the expectations of an investigation into a new application of existing methods

**Weaknesses:**

- The clarity of the writing and the organization of the paper could be significantly improved upon. I understand that the authors conducted extensive experiments and had much to discuss; however, the majority of the methodology of the paper is only described in the appendix. The main text consists mostly of lengthy sentences and very little mathematical notation, which would be instrumental in efficiently and accurately delivering concepts. If it is not possible to reorganize the paper, I believe a paper with such dense experimental results, such that the methodology does not fit within the main text, is better suited for a journal with a longer format.
- A lesser weakness is that the novelty of the paper comes from the investigated scenario, and no significantly novel methodologies are introduced.

**Questions:**

- As in line with the clarity concerns, my understanding is that throughout the experiments the authors first continue the training of the pretrained model while injecting backdoor attacks, then evaluate the model performance a): before training with the backdoor attacks b): after training with the backdoor attacks, but without inserting the triggers during evaluation, and c): while inserting the triggers. Is this correct? Regardless of the answer, I believe this should be succinctly mentioned at the beginning of the experiment section.
- Continuing on the first point, many tables essentially have the same description (e.g., 1 & 2 or 3 & 4) but show different scenarios (I believe scenarios a) and b) in the first and c) in the second). The table descriptions should be updated to accurately reflect the presented results.
- Why use the notation $\Delta_{Acc}$ while it does not represent accuracy, but rather the percentage change? I believe this notation to be confusing, especially since the description is only provided in the appendix.
- The format and contents of tables 7 & 8 are not properly explained; table 7 is not referred to in the text.
- Line 69 states: “Backdoors can generalize across unrelated tasks—poisoning one task during training can affect different tasks at inference time, **a phenomenon impossible in traditional computer vision backdoors.**” As this is a substantial claim, authors should provide evidence or citations to works containing evidence of this claim. Otherwise, I believe it should be removed from the paper.
- Line 801: Shouldn’t the mask in the loss function be inverted? For example, we want $\mathcal{L}(\tilde{x}\cdot(1 - M), x\cdot(1-M))$, as we want the pixels which were **masked in the input** to be **present in the output**.

---

> ### Author Response · Authors · 2025-11-21
>
> We value the comprehensive and insightful comments you have provided. We have addressed them below.
>
> 1. The clarity of the writing and the organization of the paper could be significantly improved upon. I understand that the authors conducted extensive experiments and had much to discuss; however, the majority of the methodology of the paper is only described in the appendix. The main text consists mostly of lengthy sentences and very little mathematical notation, which would be instrumental in efficiently and accurately delivering concepts. If it is not possible to reorganize the paper, I believe a paper with such dense experimental results, such that the methodology does not fit within the main text, is better suited for a journal with a longer format.
>
> We appreciate this concern and acknowledge that space constraints led to moving material to the appendix. We will restructure the paper, incorporating mathematical notation where appropriate and presenting the core methodology in the main text.
>
> 2. A lesser weakness is that the novelty of the paper comes from the investigated scenario, and no significantly novel methodologies are introduced.
>
> Our contribution is primarily in investigating a new threat scenario, backdoor attacks in the in-context learning setting, and demonstrating its implications. Apart from this, we also performed the first parameter backdoor attack in in-context learning. We will frame this more clearly.
>
> 3. As in line with the clarity concerns, my understanding is that throughout the experiments the authors first continue the training of the pretrained model while injecting backdoor attacks, then evaluate the model performance a): before training with the backdoor attacks b): after training with the backdoor attacks, but without inserting the triggers during evaluation, and c): while inserting the triggers. Is this correct? Regardless of the answer, I believe this should be succinctly mentioned at the beginning of the experiment section.
>
> The experimental protocol is: (a) evaluate clean pretrained model performance, (b) fine-tune with backdoor injection, (c) evaluate backdoored model on clean inputs, and (d) evaluate backdoored model with triggers present. We will explicitly state this protocol at the beginning of the experiments section.
>
> 4. Continuing on the first point, many tables essentially have the same description (e.g., 1 & 2 or 3 & 4) but show different scenarios (I believe scenarios a) and b) in the first and c) in the second). The table descriptions should be updated to accurately reflect the presented results.
>
> We will revise all table captions.
>
> 5. Why use the notation $\Delta_{acc}$ while it does not represent accuracy, but rather the percentage change? I believe this notation to be confusing, especially since the description is only provided in the appendix.
>
> Standard ASR (Attack Success Rate) applies naturally to classification tasks with discrete outputs. For our image-generation setting, we need continuous metrics to measure output degradation. We introduced $\Delta_{acc}$ to quantify this performance degradation as a measure of attack effectiveness.
>
> 6. The format and contents of tables 7 & 8 are not properly explained; table 7 is not referred to in the text.
>
> We will provide complete explanations of these tables' format and contents.
>
> 7. Line 69 states: “Backdoors can generalize across unrelated tasks—poisoning one task during training can affect different tasks at inference time, a phenomenon impossible in traditional computer vision backdoors.” As this is a substantial claim, authors should provide evidence or citations to works containing evidence of this claim. Otherwise, I believe it should be removed from the paper.
>
> The claim is based on the fact that backdoor attacks in classification are limited to outputting a wrong label. The model itself has no more ability. Take classification on CIFAR-10 as an example. The output of the model will be one of those 10 classes. However, in our case, models are much more capable; the output space is not limited to the number of classes anymore. Thus, the backdoor takes advantage of that ability. We will rewrite the text to clarify this claim.
>
> 8. Line 801: Shouldn’t the mask in the loss function be inverted?
>
> We believe there may be confusion regarding line numbering. Could you please clarify which specific formula you are referring to? We will address this once we understand the specific concern.

---

> > ### Comment · Reviewer_6VTZ · 2025-11-25
> >
> > Thank you for the explanations regarding my concerns. The authors have made several promises to incorporate some changes; however, I don't believe any changes have been made to the manuscript for this submission. Until the manuscript is updated and improvements are referenced, I am not able to change my score.
> >
> > Regarding the line confusion:
> > Line 801, referring to appendix section C.1, the first equation in the paragraph titled "Masking and Reconstruction Process". As a side note, this is another point of clarity: equations should be numbered to ease referencing.

---

### Official Review · Reviewer_EKY3 · 2025-10-30

**Soundness:** 2
**Presentation:** 2
**Contribution:** 2
**Rating:** 2
**Confidence:** 4

**Summary:**

The paper investigates backdoor attacks on ViTs trained with Masked Image Modeling and used for in-context learning in I2I tasks. The threat model exploits ViT's contextual adaptation to deploy malicious behaviors (DoS, identity mapping) on both seen and unseen tasks by adapting classical triggers (BadNets, WaNet). The backdoors achieve high attack success rates (up to 13x degradation) and are shown to be largely effective against existing defenses like fine-tuning and fine-pruning.

**Strengths:**

- The paper is the first to study backdoor attacks on ViTs trained with MIM, and provides a good background on the problem setup

**Weaknesses:**

- Although the paper is the first to study backdoor attacks for in-context learning, it relies heavily on existing attack methods and does not offer any novel theoretical insights.
- The writing needs to be thoroughly refined, as several core details (methods, metrics, and how they're formulated, background on MIM attacks, etc.) are deferred to the appendix, with a lot of redundant discussion in the main paper (mainly the results section). The methods and experiment sections talk about $\alpha$, $\delta$ and $\epsilon$ without really explaining what they denote. The main paper should be sufficiently self-contained, with only extra experiments and details provided in the appendix.
- (Minor) Most citations are formatted incorrectly.

**Questions:**

- Why and how the contextual adaptation capabilities of ViTs make them susceptible to these attacks? Rather than simply showing that the vulnerability exists, it is more valuable to examine the source

---

> ### Author Response · Authors · 2025-11-21
>
> We appreciate the insightful comments. We address each question individually:
>
> 1. Although the paper is the first to study backdoor attacks for in-context learning, it relies heavily on existing attack methods and does not offer any novel theoretical insights.
>
> We acknowledge that our work builds upon existing attack methods. However, our contribution lies in demonstrating that in-context learning introduces a new dimension of vulnerability: attackers can deploy backdoors with high flexibility, selecting different trigger designs and achieving context-dependent malicious behaviors. While our theoretical analysis is limited, we adopt an empirical approach consistent with established practices in the backdoor attack literature. We will make this positioning clearer in the revision.
>
> 2. The writing needs to be thoroughly refined, as several core details (methods, metrics, and how they're formulated, background on MIM attacks, etc.) are deferred to the appendix, with a lot of redundant discussion in the main paper (mainly the results section). The methods and experiment sections talk about ,  and  without really explaining what they denote. The main paper should be sufficiently self-contained, with only extra experiments and details provided in the appendix.
>
> We agree that the current organization has too much critical content in the appendix due to space constraints. We will restructure the main paper to be self-contained, clearly defining all notation upfront, and moving only supplementary experiments and minor details to the appendix. The core methodology, metrics, and formulations will be presented in the main text.
>
> 3. Why and how the contextual adaptation capabilities of ViTs make them susceptible to these attacks? Rather than simply showing that the vulnerability exists, it is more valuable to examine the source
>
> The fundamental vulnerability comes from MIM's reconstruction objective: the model learns to complete missing inputs. We exploit this by teaching the model to reconstruct triggers as specific malicious behaviors. When combined with in-context learning, this vulnerability becomes particularly powerful; the attacker can trigger different malicious tasks depending on the provided context.

---

### Official Review · Reviewer_q9yy · 2025-11-01

**Soundness:** 2
**Presentation:** 1
**Contribution:** 1
**Rating:** 2
**Confidence:** 4

**Summary:**

The paper shows that ViTs trained with masked-image-modeling and prompted via in-context learning can be backdoored by data poisoning so that a specific context (with a trigger) flips the model’s behavior.

**Strengths:**

1. The paper makes an interesting contribution by linking backdoor vulnerabilities to in-context learning mechanisms in vision transformers
2. The experiments cover multiple vision tasks and settings

**Weaknesses:**

1. The paper does not clearly formalize its in-context learning setup. It also remains ambiguous whether the “context” involves any language components or purely visual cues, which weakens the conceptual grounding.

2. Weak connection between MIM and in-context learning. The proposed link between masked image modeling (MIM) pretraining and in-context learning is unclear. It is difficult to see how model providers could realistically train a vision transformer using such an in-context input format.

3. The experiments are confined to a single, large Vision Transformer and a particular multi-task MIM training recipe. It is unclear whether the proposed attack generalizes to other architectures or training regimes.

4. Only prompt engineering and short fine-tuning are explored as defenses. No data sanitization, trigger synthesis search, activation clustering, spectral analysis, or parameter-space defense methods are examined. Hence, the conclusions about the difficulty of defense remain preliminary and incomplete.

5. The organization and exposition are difficult to follow, with key methodological steps scattered and insufficiently motivated. This makes the overall argumentation hard to read and interpret.

**Questions:**

Please refer to the Weakness.

---

> ### Author Response · Authors · 2025-11-21
>
> We appreciate the reviewer’s effort and time, as well as the valuable reviews. We address each question individually:
>
> 1. The paper does not clearly formalize its in-context learning setup. It also remains ambiguous whether the “context” involves any language components or purely visual cues, which weakens the conceptual grounding.
>
> We appreciate this comment. Our definitions follow the established framework from [1]. Specifically, we define in-context learning as the model's ability to make decisions (i.e., generate output images) based on contextual information without weight updates. The context consists of (source image, task image) pairs that guide the model's task selection. We will clarify these definitions in the revised manuscript.
>
> 2. Weak connection between MIM and in-context learning. The proposed link between masked image modeling (MIM) pretraining and in-context learning is unclear. It is difficult to see how model providers could realistically train a vision transformer using such an in-context input format.
>
> We acknowledge that this connection requires clearer exposition. MIM serves as the pretraining method that enables the model to reconstruct missing input parts, this is the method used in previous work [1].
> The model provider uses standard MIM pretraining with the objective of creating models capable of generalizing across diverse tasks at inference time, thereby granting the model in-context learning capabilities. We will elaborate on this relationship in the revision.
>
> 3. The experiments are confined to a single, large Vision Transformer and a particular multi-task MIM training recipe. It is unclear whether the proposed attack generalizes to other architectures or training regimes.
>
> We clarify that our experiments include multiple Vision Transformer variants (small, medium, and large). Furthermore, our method is architecture-agnostic and its extension to other ViT-based models is natural. We will emphasize this generalizability in the revised paper. We argue that as long as the models are capable of the in-context learning capability, they could be affected by our attack.
>
> 4. Only prompt engineering and short fine-tuning are explored as defenses. No data sanitization, trigger synthesis search, activation clustering, spectral analysis, or parameter-space defense methods are examined. Hence, the conclusions about the difficulty of defense remain preliminary and incomplete.
>
> Our primary contribution focuses on demonstrating that triggers can launch different backdoor behaviors conditioned on context, rather than on novel trigger design. By evaluating diverse trigger types, we show that our approach generalizes across trigger families, which an attacker could leverage to circumvent various defenses. Consequently, evaluating specific advanced defenses (data sanitization, trigger synthesis search, etc.) would yield results heavily dependent on trigger design choices rather than showing fundamental vulnerabilities in the in-context learning paradigm. We will clarify this scope in the revision.
>
> 5. The organization and exposition are difficult to follow, with key methodological steps scattered and insufficiently motivated. This makes the overall argumentation hard to read and interpret.
>
> We appreciate this feedback and will revise the manuscript to improve clarity, better motivate methodological steps, and enhance overall readability.
>
> References
>
> [1] Wang, Xinlong, et al. "Images speak in images: A generalist painter for in-context visual learning." Proceedings of the IEEE/CVF Conference on Computer Vision and Pattern Recognition. 2023.

---

### Author Response · Authors · 2025-12-03

Dear Reviewers,

We sincerely thank you for your thorough and constructive feedback. We have carefully addressed all the comments and concerns raised in the previous round of reviews. We have uploaded a new version of the manuscript with all changes highlighted in blue for easy identification.
Below, we summarize the major revisions made to address each reviewer's concerns:

Summary of changes:

1. (Reviewer q9yy) Formalization of In-Context Learning Setup in Section 2.1
2. (Reviewer q9yy) MIM and In-Context Learning Connection in Section 2.1
3. (Reviewers EKY3 & 6VTZ) Notation and Mathematical Clarity in Section 4
4. (Reviewer 6VTZ) Experimental Protocol Clarification in Section 5
5. (Reviewer 6VTZ) Mask Formula Correction
6. (Reviewer KJ79) Defense Discussion in Section 5.4
7. (Reviewer KJ79) Threat Model Expansion in Section 3
8. (Reviewer 6VTZ) Improved Table Descriptions and References

General changes:
Novelty concerns (Reviewers EKY3, 6VTZ, KJ79):
We have clarified that our contribution is primarily empirical, focusing on revealing a previously unexplored threat surface. We have also emphasized our parameter-space backdoor attack as a novel technical contribution in this setting.

Organization concerns (All reviewers):
We have restructured the paper to be self-contained within the main text, with notation defined upfront and the methodology clearly presented before the results.

Thank you again for your valuable feedback.

---

### Meta-Review · Area_Chair_Ei9A · 2026-01-01

**Summary:**

This paper investigates backdoor attacks against Vision Transformers under in-context learning for image-to-image tasks, identifying in-context visual adaptation as a new attack surface and empirically demonstrating flexible, context-dependent backdoor behaviors across tasks, including unseen ones. All reviewers agree that the problem setting is timely and that the experimental study is extensive, covering multiple trigger types, attack objectives, and task-specific versus task-agnostic regimes. However, the reviewers consistently raised concerns about limited methodological novelty, heavy reliance on existing trigger mechanisms, and significant clarity and organization issues, particularly the placement of core methodology and notation in the appendix rather than the main text. In their responses, the authors provided reasonable explanations, clarified the purely visual in-context learning setup, strengthened the connection between masked image modeling and contextual adaptation, corrected technical issues, and committed to substantial restructuring to improve readability and self-containment. Had a full discussion phase occurred and a revised manuscript been evaluated, these clarifications would likely have improved perceptions of soundness and presentation, and one reviewer explicitly raised their score after the rebuttal; nevertheless, the core concern regarding incremental novelty relative to prior backdoor work would likely persist for other reviewers, with scores at best moving to a borderline range. Overall, while the work offers a valuable empirical exploration of an underexplored threat scenario and may stimulate further research on securing in-context vision models, its current contribution is primarily investigative rather than introducing a new attack formulation or defense, and the clarity issues remain unresolved at submission time. Based on these considerations, I judge that the paper does not yet meet the bar for acceptance at ICLR, but could become more competitive with a substantially revised, more focused presentation and a clearer articulation of technical novelty.

**Reviewer Concerns:**

Across all reviews, there was broad agreement that the paper identifies a relevant and timely problem, i.e., backdoor vulnerabilities in in-context learning for vision transformers, and that the experimental evaluation is extensive across multiple tasks, triggers, and attack settings. Several concerns raised by reviewers were at least partially addressed in the rebuttal. In particular, the authors clarified that the in-context learning setup is purely visual (without language components), strengthened the explanation of how masked image modeling enables contextual task adaptation, corrected technical and notation issues, and provided clearer descriptions of the experimental protocol and evaluation metrics. One reviewer explicitly acknowledged that these responses resolved their main concerns.

However, multiple substantive concerns remain outstanding. Most importantly, reviewers consistently noted that the technical novelty is limited, as the work primarily adapts existing backdoor trigger mechanisms and attack objectives to a new setting without introducing a fundamentally new attack formulation, training objective, or defense. In addition, while the authors committed to reorganizing the paper and moving core methodology and notation into the main text, these improvements are not reflected in the submitted version to a large extent, leaving significant clarity and presentation issues unresolved at the time of evaluation. Finally, the defense analysis remains limited to relatively simple baselines, which constrains the generality of the conclusions regarding the ineffectiveness of defenses. These unresolved issues collectively informed the final recommendation.

**Reviewer Scores:**

Based on the rebuttal and limited discussion, I believe that reviewer scores would have changed only modestly if a full discussion period had occurred. For reviewers primarily concerned with clarity and problem formulation, the authors’ explanations would likely have led to a slight increase in perceived soundness and presentation, potentially moving their scores from clear reject to borderline reject. This is supported by the fact that one reviewer explicitly raised their score after the rebuttal.

However, for reviewers whose main concern was the lack of methodological novelty and the incremental nature of the contribution relative to prior backdoor literature, these concerns were largely acknowledged rather than resolved. As a result, their scores would likely have remained unchanged, or at best increased marginally, and consensus toward acceptance would still be unlikely. Overall, even under a full discussion scenario, the paper would most plausibly have remained in the borderline-to-reject range rather than converging to a clear accept.

---

### Decision · Program_Chairs · 2026-01-26

Reject